# Porous organic cages as synthetic water channels

Yi Di Yuan[1], Jinqiao Dong[1], Jie Liu[1], Daohui Zhao[1], Hui Wu [2], Wei Zhou [2], Hui Xian Gan[1,3],
Yen Wah Tong [1,3], Jianwen Jiang [1✉] & Dan Zhao [1✉]

Nature has protein channels (e.g., aquaporins) that preferentially transport water molecules while rejecting even the smallest hydrated ions. Aspirations to create robust synthetic counterparts have led to the development of a few one-dimensional channels. However, replicating the performance of the protein channels in these synthetic water channels remains a challenge. In addition, the dimensionality of the synthetic water channels also imposes engineering difficulties to align them in membranes. Here we show that zero-dimensional porous organic cages (POCs) with nanoscale pores can effectively reject small cations and anions while allowing fast water permeation (ca. $10^9$ water molecules per second) on the same magnitude as that of aquaporins. Water molecules are found to preferentially flow in single-file, branched chains within the POCs. This work widens the choice of water channel morphologies for water desalination applications.

[1] Department of Chemical and Biomolecular Engineering, National University of Singapore, 4 Engineering Drive 4, 117585 Singapore, Singapore. [2] NIST Center for Neutron Research, National Institute of Standards and Technology, Gaithersburg, MD 20899-6102, USA. [3] National University of Singapore, NUS Environmental Research Institute (NERI), 117411 Singapore, Singapore. ✉email: chejj@nus.edu.sg; chezhao@nus.edu.sg

Seawater desalination is key to alleviate the escalating global freshwater scarcity[1]. The discovery of aquaporins' surprisingly high water permeability (ca. $3 \times 10^9$ water molecules per second per channel) and perfect desalting ability due to the subnanometer-sized (ca. 2.8 Å diameter) pore channels has inspired studies to acquire them for enhancing current water desalination techniques[2–4]. Using aquaporin as a benchmark, synthetic efforts to mimic the functional properties of aquaporins with added robustness have afforded multiple analogs[5–13].

Bioinspired synthetic water channels commonly have water permeable central apertures surrounded by hydrophobic outer shells for stabilizing within the hydrophobic core of lipid bilayer systems[14]. The two general synthetic strategies are unimolecular tubular architectures[5–9,13] and bottom-up assemblies[10–12]. Until recently, the highest water permeation of synthetic water channels was reported in unimolecular channels, e.g., 0.8-nm-diameter carbon nanotubes with water permeabilities about six times higher than that of aquaporins[9]. However, the window opening sizes of most unimolecular channels are still too large for complete salt rejection. As highlighted by Patel et al.[15], purely increasing water permeability will only marginally reduce specific energy consumption. Increasing water-solute selectivity, i.e., improving salt rejection, would be more effective at improving energy efficiency. Hence, water channels with both high water permeability and low or negligible ion permeation are favored. Empirically observed, channels with a window opening size of ca. 3 Å (close to that of aquaporins) can effectively exclude hydrated ions[11–13]. Unfortunately, this may come at the expense of much lower water permeabilities[12]. Recently, Song et al.[13] reported that both high water permeation on the scale of aquaporin and salt rejection can be achieved in a ca. 3 Å unimolecular channel through planar clustering of the channel, where water preferentially flows through larger side openings. This strategy imposes a critical vertical alignment configuration such that exposing the larger channel sides to the salt-rich feed can potentially lower its salt-rejection efficiency. Such alignment issues in unimolecular channels and conformation stability of the self-assembled channels[16] can significantly impede the channel performance. These factors represent a formidable design challenge in synthetic chemistry.

In this study, we report porous organic cages (POCs) as orientation-independent synthetic water channels with both high water permeability and salt rejection. POCs are a class of discrete molecules with synthetically tunable window opening size and functionality that are fully organic in construct[17]. Unlike other advanced porous materials such as metal–organic frameworks (MOF) or covalent organic frameworks (COFs) that occur as frameworks (three-dimensional, 3D), sheets (two-dimensional, 2D), and rods (one-dimensional, 1D), molecular cages can dissolve and exist as a single molecular entity (zero-dimensional, 0D). Interestingly, most of the POCs have good structural symmetry with windows on many sides leading to the internal cavity despite possible random rotations[18]. Here we choose a class of tetrahedral-shaped POCs containing four triangular windows leading to a central cavity (Fig. 1a). Most of such POCs can align window-to-window to form extended 3D pore networks consisting of internal cavities within each POC and external cavities between POCs where guest molecules can traverse (Fig. 1b, c) irrespective of the orientation of the POCs. Molecular simulations of water desalination using bulk tetrahedral POC membranes or POCs in lipid bilayer have shown good water permeation and full salt rejection[19–21]. Previous studies have also indicated that water molecules can reversibly reside within the cavity of POCs and their 3D pore networks, enhancing protonic conduction as compared to one-dimensional channels[22,23]. Thereby, here we explore the efficacy of tetrahedral POCs as synthetic water channels and elucidate the structure–performance correlation through experimental and simulation studies.

## Results

**Insertion of POCs into a lipid bilayer.** We chose six tetrahedral POCs namely CC1, CC3, RCC3, FT-RCC3, CC5, and CC19 (Fig. 1a and Supplementary Fig. 1) to systematically study four factors (i.e., window opening size, structural stability, pore network connectivity, and hydrophilicity) that may influence the performance of POCs as water channels[24–27]. The water permeability of water channels is typically investigated using liposome shrinkage or swelling tests with a stopped-flow light-scattering apparatus[8]. Ion permeability, on the other hand, is commonly investigated using fluorescence spectroscopy techniques[28]. All these techniques require water channels to be inserted into bilayer systems such as planar lipid bilayers and liposomes. In this study, we embedded POCs into liposomes using the reverse-phase method as POCs can only be dissolved in organic solvents[29]. The successful incorporation of POCs within the lipid bilayer was verified with fluorescence confocal spectroscopy (Fig. 1f, g and Supplementary Fig. 2), cryogenic transmission electron microscopy (cryo-TEM, Supplementary Fig. 3), Fourier-transform infrared spectroscopy (FTIR, Supplementary Fig. 4), and ultraviolet-visible spectrophotometry (UV-Vis, Supplementary Fig. 5). Two POCs, CC19, and CC5 are highly fluorescent while the lipids used in the experiments are non-fluorescent. The appearance of fluorescent rings is evident that the POCs have been preferentially encapsulated within the lipid bilayer. No irregularities were observed on the liposomes, suggesting that the incorporation of POCs into lipid bilayer did not affect the bilayer formation. To increase the visual contrast of the wholly organic POCs under cryo-TEM, palladium nanoclusters were encapsulated within RCC3 using the reported method (Supplementary Figs. 6 and 7)[30]. The presence of darkened objects within the lipid bilayer suggests that POCs are nanometer-scale in the lipid bilayer, which in turn controls the size of POC nanoaggregates within the bilayer thickness (ca. 5 nm). Each POC has a diameter of ca. 2 nm, which is too small to transverse the lipid bilayer. In order to prove this, we simulated an aggregate of POC containing three CC3 molecules in the lipid bilayer (Supplementary Movie 1) and observed no water permeation through the POC aggregate. Therefore, we expect the POCs to form ca. 5 nm transmembrane nanoaggregates (Supplementary Fig. 8) with short-range molecular ordering which is possible considering that symmetrical cages have a high propensity to crystallize[31]. Under liquid and solid atomic force microscopy (AFM), supported lipid bilayer (SLB) incorporated with CC3, formed by rupturing liposomes over solid support, appears to be rougher (Ra = 0.657 nm) compared to the blank lipid bilayer (Ra = 0.279 nm) while no obvious protrusion was observed (Supplementary Fig. 9). This suggests that the CC3 nanoaggregates inserted into the bilayer may be similar in size to the bilayer thickness.

**Factors affecting water permeation through POCs.** POCs of increasing feed molar ratios of POCs over lipids (normally referred to as feed molar channel/lipid ratio, fmCLR) were embedded into liposome to test their water and salt permeabilities (Fig. 2). Notably, herein fmCLR describes the initial sample preparation ratio, with the mole of POCs introduced over the mole of lipids used and is not corrected with the actual embedding efficiency. During the water permeability measurement, liposome shrinkage was induced under rapid exposure to a hypertonic buffer solution containing sucrose osmolyte. The light-scattering signal at 90° increased and the resultant curve was fitted with a double-exponential function describing two shrinkage rates, $k_1$ (permeation through lipid bilayer) and $k_2$

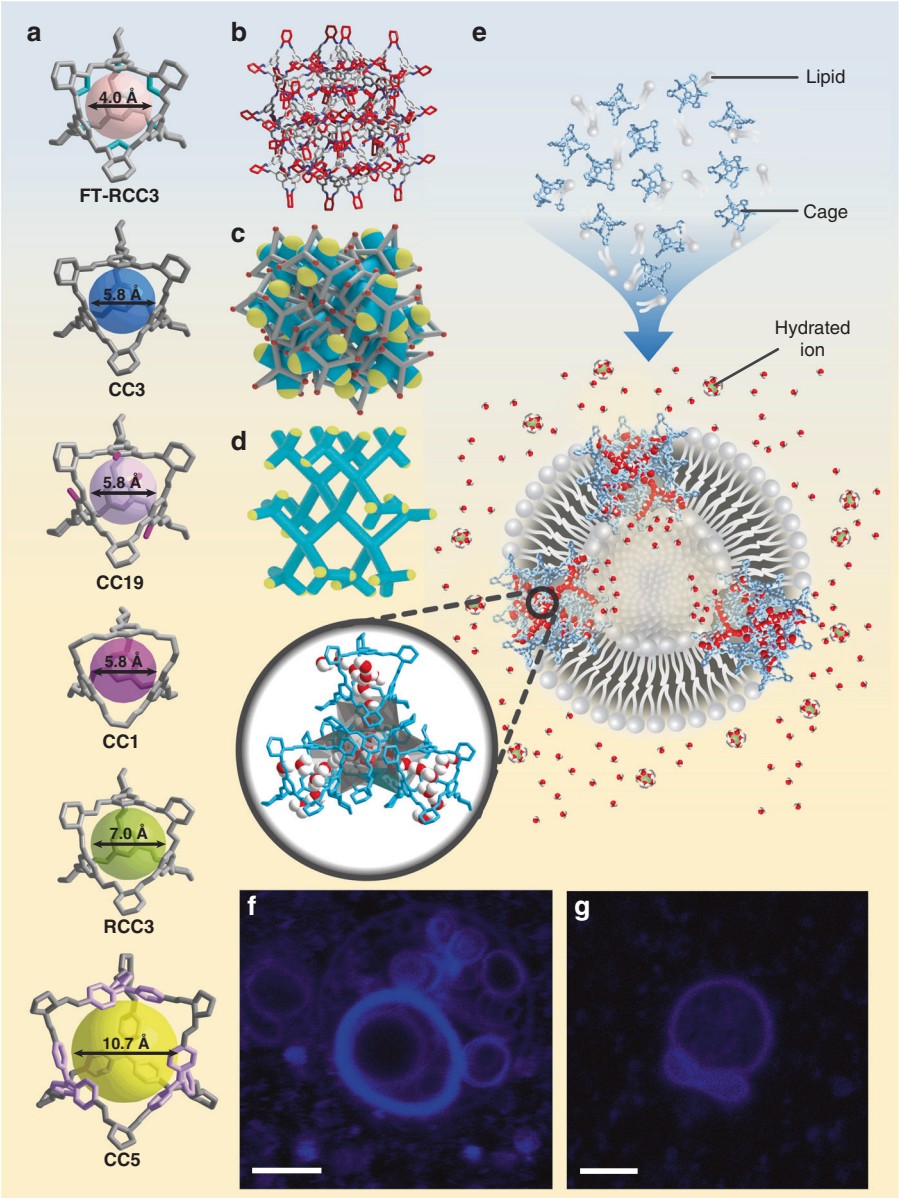

**Fig. 1 POCs and POC nanoaggregates in liposome. a** Crystal structures of FT-RCC3, CC3, CC19, CC1, RCC3, and CC5. **b** Structure of window-to-window packing of tetrahedral POCs. **c** Side view of a tetrahedral POC crystal (gray with red vertex group) with channel network (blue) shown. Ends of the channel network (yellow) are possible entry points for guest molecules. **d** Structure of an extended channel network without tetrahedral POC shell. The tetrahedral 3D channels run through cage cavities and inter-cage gaps with the node at the center of the cage cavity. **e** Scheme of CC3 nanoaggregates (light blue) in lipid bilayer with water chains formed inside the channels. Insert: CC3 molecules shown in blue with the water channel illustrated in gray. Fluorescence confocal microscopy of liposomes with CC5 (**f**) and CC19 (**g**) under an excitation of 402 nm laser. Blue circles indicate the presence of CC5 and CC19 in the lipid bilayer. Scale bar represents 5 μm.

(permeation through POCs)[13]. We extracted the water permeability data from $k_2$ (Supplementary Data 1). The water permeabilities of CC3, CC5, CC19, and FT-RCC3 increased with increasing fmCLR until reaching maximum values and plateauing thereafter (CC19 and FT-RCC3 plateau after fmCLR of 0.05, Supplementary Fig. 10a and Supplementary Tables 1 and 2). The maximum water permeabilities through CC3, CC5, CC19, and FT-RCC3 were found to be 359 (±63.2), 389 (±49.9), 291 (±37.7), and 340 (±46.8) μm s$^{-1}$, respectively. Pohl and co-workers recently presented an updated model to calculate the water permeabilities[32,33]. Using the new model, the corrected water permeabilities were calculated as 135 (±23.7), 146 (±18.7), 109 (±14.1), and 127 (±17.6) μm s$^{-1}$ for CC3, CC5, CC19, and FT-RCC3, respectively, which are smaller than the permeabilities

based on a conventional model but on the same magnitude. It is worth noting that the water permeabilities of the POCs are higher than that of most synthetic water channels under shrinkage test conditions, such as peptide-appended pillar[5]arene (1 μm s$^{-1}$)[8], I-quartet (3–4 μm s$^{-1}$)[12], etc.

The water transport though water channels largely depends on the dimension of channels' smallest constrictions. In our case, the smallest constrictions occur at the windows of the POCs. Thus, we compared CC3, CC5, and FT-RCC3 for the effect of window size on water transport. The window opening size increases in the order of FT-RCC3 (4.0 Å) < CC3 (5.8 Å) < CC5 (10.7 Å)[26,27]. This trend is reflected in the fmCLR of POCs to reach maximum water permeabilities. Smaller loadings of POC in liposome can be used to achieve maximum water permeabilities for POCs with larger

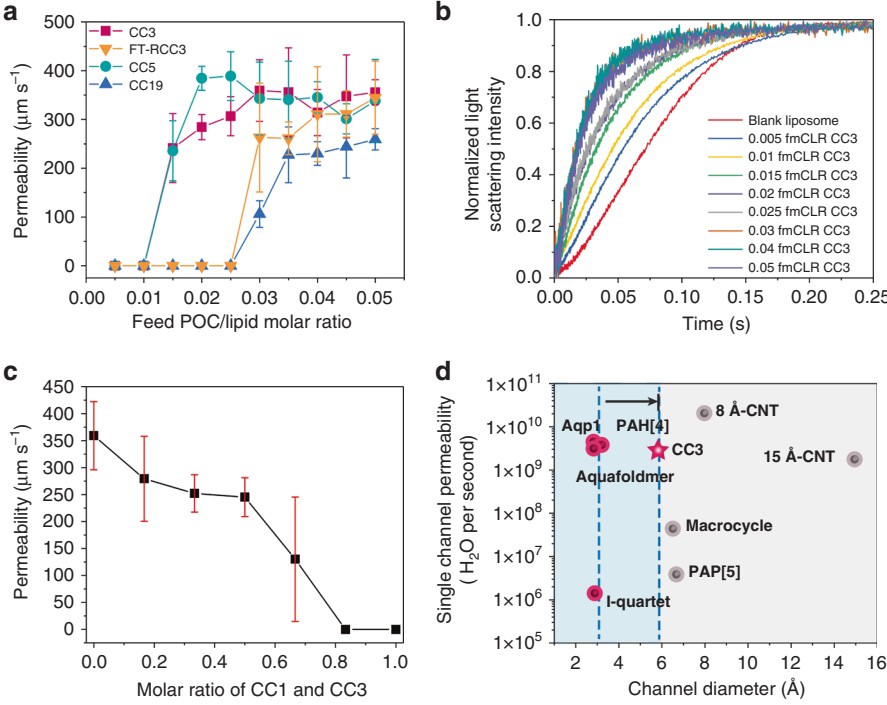

**Fig. 2 Water permeability of liposomes embedded with POCs. a** Permeabilities of POCs (CC3, FT-RCC3, CC5, and CC19) with loadings from 0 to 0.05 fmCLR measured under hypertonic conditions at 25 °C. **b** Representative stopped-flow traces from tests performed on liposomes embedded with CC3 of various fmCLRs. **c** Water permeabilities of liposomes with a combined fmCLR of 0.03 for CC1/CC3 mixture. Molar ratio of 0.0 refers to fmCLR of 0.03 for CC3 (no addition of CC1), and molar ratio of 1.0 refers to fmCLR of 0.03 for CC1 (no addition of CC3). **d** Single-channel water permeabilities of CC3 (this study) and synthetic channels with reported stopped-flow values[4,8–13]. Pink color indicates channels with total ion rejection while gray color indicates channels with partial to no cation rejection. Notably, the CC3 water channel can push the limit of ion-rejecting channel diameter from 2.8 to 5.8 Å, as indicated by the arrow. Error bars represent standard deviations of three independent replicates.

window sizes (Supplementary Table 1). In contrast, water permeation through liposomes with RCC3 or CC1 resembles that of a blank liposome (Supplementary Fig. 10c, d). This observation suggests that the aggregates of these POCs may be nonporous within the lipid bilayer. It is well known that RCC3 has flexible amine bonds that can twist and collapse the cage when the cage is desolvated, resulting in a smaller or non-existent internal cavity[26]. Therefore, RCC3 may remain in a collapsed and nonporous structure inside the liposome, and water transport is only possible through the lipid bilayer. Although CC1 has similar window opening size and rigidity as that of CC3, it differs from CC3 in its packing mode where the windows of each cage molecule are blocked by the arenes of its neighbors[24]. This prevents the formation of an interconnected pore network within the CC1 packing despite still having both internal and external cavities. To further verify the effect of arene-to-window packing on water permeation through POCs, we added CC1 to CC3 in increasing ratios while fixing the total fmCLR at 0.03. Water permeability decreases with the increasing ratio of CC1 until no channel permeation was observed (Fig. 2c). The lack of water permeation through CC1 suggests that an interconnected pore network system is present and necessary in a tetrahedral POC system within the lipid bilayer to permit water transport. In short, structural rigidity is crucial for POCs as the building units for water channels, and interconnected pore networks are necessary for POC aggregates to traverse lipid bilayers for water transport.

Interestingly, the performance of CC1 provides evidence that the POCs may exist as crystalline structures within the lipid bilayer. CC1, despite exhibiting an apparent lack of porosity when in crystalline state[19], may contain some interconnected networks when the molecules pack in an amorphous fashion[20]. Therefore, some water permeation may be expected if CC1 packs

amorphously. To further explore the effect of amorphous packing on the water permeability of POCs, amorphous scrambled POC (ASPOC) mixing CC1 and CC3 ligands were tested for its water permeability[31]. ASPOC of mass similar to that of 0.03 fmCLR CC3 was added to the liposome, and no significant channel permeation was observed (Supplementary Fig. 10e). This suggests that the POCs may be crystalline inside the lipid bilayer and the interconnected pore network facilitates water permeation.

Besides the effect of pores, the hydrophilicity of POCs (Supplementary Fig. 11) also plays an important role in controlling water transport. For example, water transport seems to be more difficult in FT-RCC3 as we only observed water permeation after 0.025 fmCLR. Threshold at 0.01 fmCLR was also observed for CC3 and CC5, which is a result of cooperative effect such that sufficient concentration of POC is required to form nanoaggregates containing transmembrane channels. The further delay of the threshold in FT-RCC3 can be attributed to the hydrophobicity contributed by the methylene groups in its cavity. Hence, more channels are needed to reach the maximum water permeability. We observed another delayed trend in CC19, which is a more hydrophilic POC. CC19 has a structure similar to that of CC3, but with zero to three hydroxyl groups lining its windows that are capable of forming hydrogen bonds. We observed similar water arrangement in CC3 and CC19 crystals immersed in deuterium oxide (D$_2$O) at room temperature using neutron powder diffraction measurements (Fig. 3a, b, Supplementary Fig. 12). Due to the higher hydrophilicity of CC19 compared to that of CC3, CC19 is able to encapsulate more D$_2$O in its cavity and holds a more complex hydrogen-bonding system. The occupancy of the D$_2$O located next to the hydroxyl groups in CC19 is 100%, notably higher than that of CC3 (ca. 70% at identical positions). It is apparent that hydrophilicity can

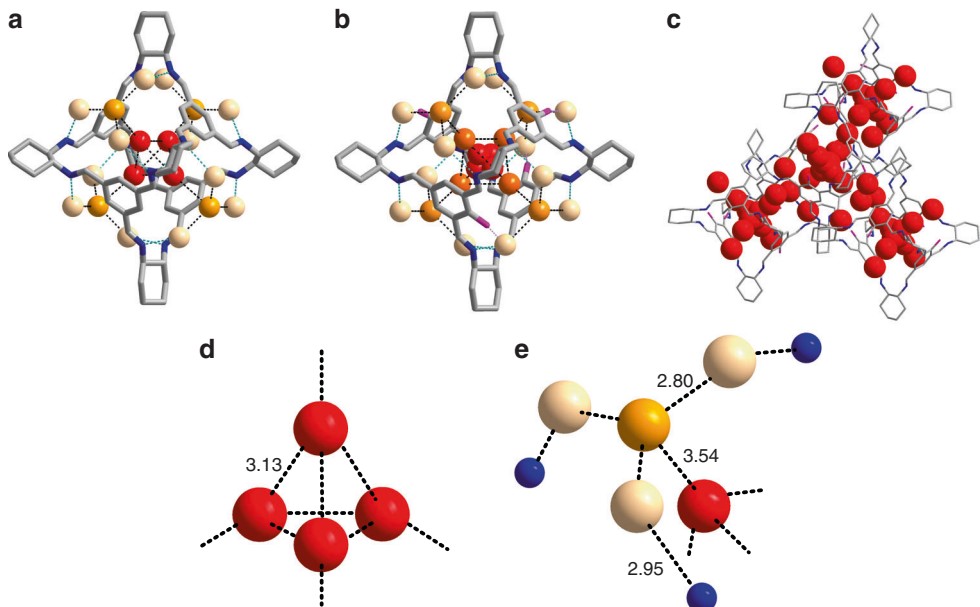

**Fig. 3 Water molecule distribution in POCs.** Structures of $D_2O$ in CC3 (**a**) and CC19 (**b**), determined from neutron powder diffraction measurements. Carbon, gray; nitrogen, blue; hydroxyl oxygen, pink. The oxygen atoms from deuterium oxide are presented in balls with various colors (i.e., red, orange, yellow, and light yellow) to indicate crystallographically independent $D_2O$ species. Deuterium and hydrogen atoms are omitted for clarity. **c** Arrangement of oxygen atoms (red) of $D_2O$ in four closely packed CC19 molecules. Peripheral oxygen atoms are omitted for clarity. Crystallographic arrangement of $D_2O$ in the internal cavity (**d**) and inter-POC space (**e**) of CC3 with possible hydrogen bonding between the $D_2O$ molecules and the POC. Nitrogen atoms from CC3 are represented as blue spheres. Note that the orientational disorder of $D_2O$ was expected at room temperature, and the $D_2O$ molecules were modeled with multiple deuterium sites.

significantly increase the capacity of POCs to attract and store water molecules. However, water movement is likely to be limited by the time taken for the sequence of hydrogen bond breakage, reorientation of water molecules, and bond reformation[32]. Therefore, despite having a similar window and cavity size, greater loading of CC19 in the liposome is needed to compensate for the higher resistance in water transport.

**Ion permeation through POCs.** Ion rejection is another important parameter in evaluating the feasibility of water channels for desalination. A POC has been previously reported to fully reject cations but preferentially allow high iodide permeation[34]. The tetrahedral POCs in this study, however, showed both negligible cation and anion permeation (Supplementary Fig. 13). Although the negligible difference in chloride and bromide ions was observed for liposomes with and without POCs, iodide permeation through the liposomes with POCs was observed to be smaller compared to that of blank liposomes (Supplementary Fig. 13l). For the hydrophobic POCs with windows smaller than 7 Å, the entrance of hydrated ions into POCs may be restricted by the dehydration energy penalty arising from the absence of surrogation in the hydrophobic structures for the water of hydration[32]. Furthermore, because of the interconnected pore network within the POC system, further energy penalty may incur due to the repeated hydration of ions in the POC cavities and dehydration at the narrower windows as the ions traverse from one POC to the next in the nanoaggregate. This may explain the lack of ion permeation through CC19 despite having hydroxyl groups that may surrogate and encourage the dehydration of hydrated ions.

**Computational simulation of water and ion permeation through POCs.** To provide microscopic insights into the size and

hydrophilicity effects of POCs on water permeation, molecular dynamics (MD) simulations were performed to investigate water permeation through POC-embedded lipid membranes. At equilibration, the thickness of the pure lipid membrane was calculated to be $3.87 \pm 0.1$ nm. Considering our observation that the nanoaggregates of POCs should traverse the bilayer with good channel connectivity, we inserted a nanocrystal consisting of 17 molecules of CC3 or CC19 (17-POC) into the lipid layer for the simulation. Instead, a larger 75-POC nanocrystal (i.e., 75 molecules of POC per nanocrystal) was used for CC5 as we observed the tendency of lipid tail invasion that can rapidly block the pores of the 17-POC nanocrystal of CC5 during the simulation (Supplementary Movie 2 and Supplementary Fig. 8d). The low root-mean-square deviations (RMSD) indicate that the POC structures were well maintained in the lipid membrane during simulation without significant structural deformation (Supplementary Fig. 8c). This is largely attributed to the like-like match between the hydrophobic outer surface of POCs and the hydrophobic domain of the lipids.

On the basis of the simulation results, the water molecules inside CC3 and CC19 nanoaggregates can form single-file chains at every branch before meeting another chain at a node inside the internal cavity of a POC molecule (Fig. 4a). This resembles the arrangements of $D_2O$ molecules in POCs (Fig. 3c). Translocation of water molecules as singly aligned waterwires has been reported in typical natural and synthetic water channels which are capable of salt rejection[9,11–13,35,36]. This is because of the confinement of water within channels with window opening sizes smaller than 1 nm. Along the short single-file chain, all the hydrogen bonds point in the same direction to form an orientationally and dipolarly ordered arrangement of water molecules. Such an orientation is preserved even at the nodes, rendering the neighboring chains directed at opposite orientations. However, single-file water chains were not observed in CC5. Instead, its

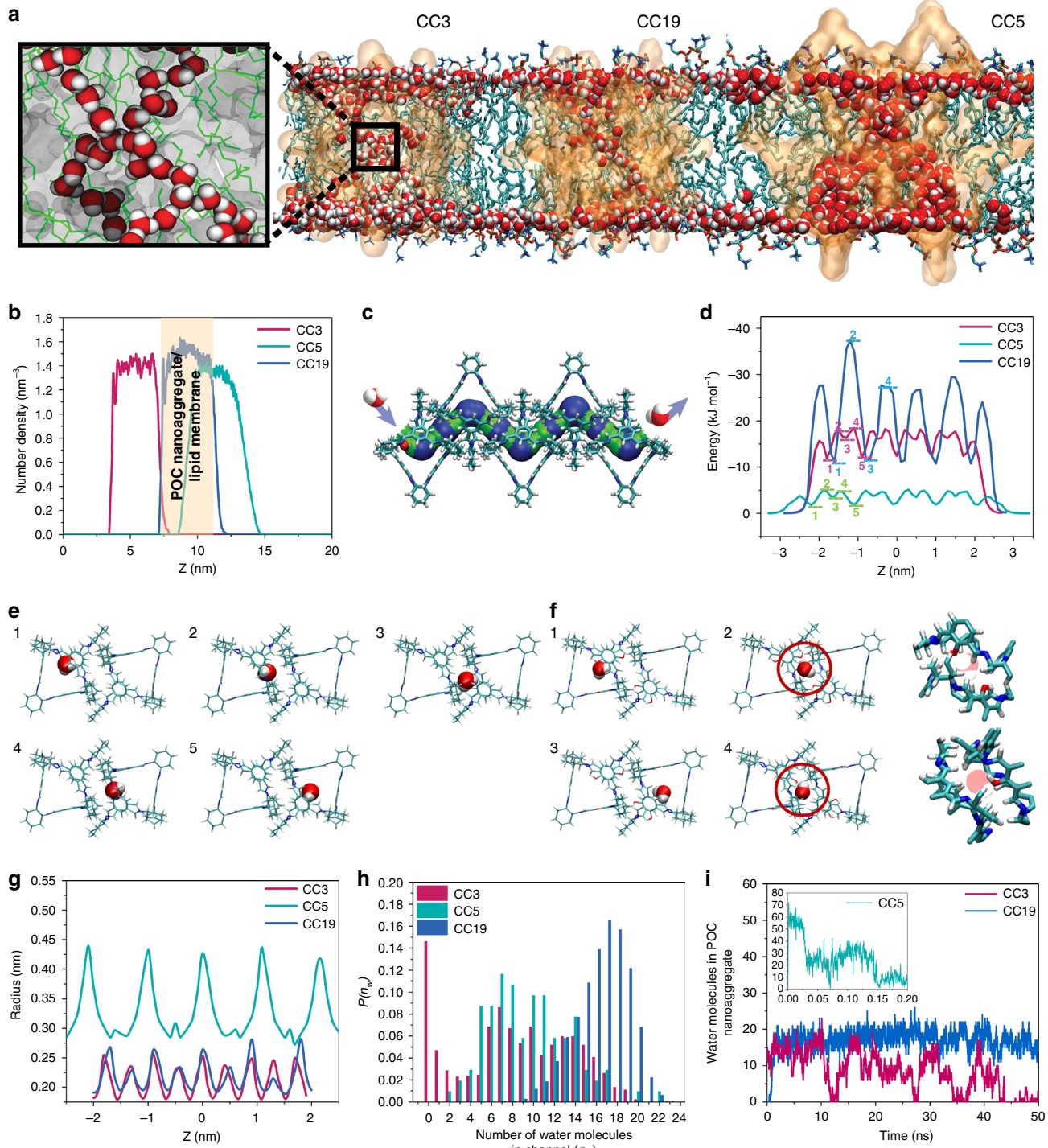

**Fig. 4 Molecular simulation of water permeation through POC nanoaggregates embedded within lipid bilayer. a** Simulation of water chains formed within the CC3, CC19, and CC5 nanoaggregates embedded within the POPC lipid bilayer. Insert: enlarged water chains inside CC3 nanoaggregate (in transparent gray). **b** Number density of Na⁺ ions through the lipid bilayer embedded with various POC nanoaggregates. **c** Scheme of water flow through a POC channel. **d** Interaction energies of water molecules with CC3, CC19, and CC5 channels. Water molecules interact with CC3 and CC5 through a 5-step mechanism (**e**), while through a 4-step mechanism with CC19 because of the presence of hydrophilic hydroxyl groups in CC19 (**f**). **g** Channel radii of CC3, CC19, and CC5. **h** Water occupancy ($n_w$) probability in CC3, CC19, and CC5 channels. **i** Wetting and dewetting patterns of CC3, CC19, and CC5 (insert).

large pore and cavity size allow water to behave like a bulk state. The cation and anion rejections of CC3 and CC19 are predicted to be 100% (Fig. 4b and Supplementary Table 3), which is consistent with the above experimental results and the previous simulation studies on the crystalline and amorphous membranes of CC3[19–21]. Despite its large pore windows, CC5 only showed

slight to negligible cation permeation, which is consistent with the experimental observation.

To qualitatively understand the water permeation, we evaluated the binding energy profiles of single water translocation calculated from potential energies as well as the wetting–dewetting transitions in the POC channels. Ascribed to the geometric effect, the

minimum energy occurs in the middle of the cage cavity, whereas the maximum energy takes place between the windows of two cages. Hence, moving through the POCs, water molecules experience the strongest energy barrier at positions 2 and 4 where they move across the narrower windows (Fig. 4d−f). Intuitively, the energy profiles of CC3 and CC5 show symmetrical patterns due to their symmetric structures, whereas the pattern of CC19 is asymmetrical due to the presence of zero to three hydroxyl groups at the pore windows. As shown in Fig. 4d, the energy at position 2 of CC19 is stronger than that at position 4. This is because of the presence of two hydroxyl groups at position 2 while only one at position 4 (Fig. 4f). Consequently, CC19 ($26.33 \, kJ \, mol^{-1}$) has a much higher energy barrier (between positions 1 and 2) than that of CC3 ($6.04 \, kJ \, mol^{-1}$) due to the affinity between water and the hydroxyl groups. Among the 3 POCs, CC5 ($3.16 \, kJ \, mol^{-1}$) has the lowest water energy barrier because of its large window opening size (Fig. 4g).

Wetting–dewetting transitions were observed in some of the reported water channels[5,8]. It represents two hydration states in which the water channel is either occupied or empty. Frequent episodes of dewetting can significantly impede water permeation. Figure 4h shows the occupancy probability $P(n_w)$ of the number of water molecules $n_w$ in the POC channels. The $n_w$ was counted within the middle segment of the channel (1 nm height). Intriguingly, in CC3, the maximum $P(n_w)$ is achieved when $n_w = 0$, indicating very frequent wetting–dewetting transitions in its channels (Fig. 4i and Supplementary Movie 3 and Supplementary Fig. 8h). Whereas for CC19, the channels are always filled with water during the simulation (Fig. 4h, i and Supplementary Movie 4 and Supplementary Fig. 8i). Its $n_w$ oscillates between 9 and 23, and the maximum $P(n_w)$ occurs at $n_w = 17$. The absence of channel dewetting in CC19 can be attributed to the higher hydrophilicity conferred by its hydroxyl groups that encourages water–channel interactions. In CC5, the wetting–dewetting transition is also absent due to its large porosity that allows bulk-like water transport (Supplementary Movie 5 and Supplementary Fig. 8j).

The simulated water permeabilities through the 17-POC nanocrystals of CC3 and CC19 are $9.4 \times 10^8$ and $8.6 \times 10^8$ $H_2O$ per second, respectively. To compare with the simulated values, we converted the experimental water permeabilities to single-nanoaggregate permeabilities where each nanoaggregate was assumed to be a 17-POC nanocrystal. At the maximum total channel permeabilities, single nanoaggregates of CC3 and CC19 have water permeabilities of $2.85 \, (\pm0.50) \times 10^9$ and $1.27 \, (\pm0.16) \times 10^9$ $H_2O$ per second, respectively. Using the updated model proposed by Pohl and co-workers[32,33], the corrected single-nanoaggregate permeabilities were determined to be $1.07 \, (\pm0.19) \times 10^9$ and $4.80 \, (\pm0.61) \times 10^8$ $H_2O$ per second for CC3 and CC19, respectively. Interestingly, the corrected water permeabilities are in better agreement with the simulated results. The good agreement suggests that CC3 and CC19 may exist as aggregates containing around 17 cage molecules per aggregate in the lipid bilayer. The water permeability of the 75-POC nanocrystal of CC5 was predicted to be $2.82 \times 10^{10}$ $H_2O$ per second. Hence, we assumed a 75-POC nanocrystal for CC5 and estimated its single nanoaggregate water permeability to be $1.60 \, (\pm0.21) \times 10^{10}$ $H_2O$ per second (the corrected permeability is $6.02 \, (\pm0.77) \times 10^9$ $H_2O$ per second). Notably, the single nanoaggregate permeabilities of some of the POCs in this study are on the same order of magnitude as that of aquaporins (see CC3 in Fig. 2d), suggesting an exciting direction for synthetic water channels.

In summary, we have demonstrated the efficacy of POCs as perfectly selective water channels using experimental methods. The water and ion permeabilities of the POCs are in good agreement with the MD simulation results. We found that the

POCs' pore window size, structural rigidity, hydrophilicity, and their ability to form interconnected channel networks are the major factors determining their water and ion permeation. The highly symmetric structures of POCs present an excellent opportunity to develop highly efficient and orientation-free synthetic water channels. These solution-processable molecules can potentially be homogeneously processed into composite materials such as membranes using facile engineering methods for desalination applications. Considering their easily tunable window size and chemical nature, POCs are possible candidates for more directed and precise water separations such as solute–solute separation that can minimize desalination post-treatment[37]. Furthermore, the scaling-up ability of POC synthesis[38] can markedly reduce the materials cost for larger-scale applications.

## Methods

**POC reconstitution into lipid vesicles using reverse-phase method**. In general, the liposome samples were prepared using 1,2-dioleoyl-sn-glycero-3-phosphocholine (DOPC) or egg-yolk phosphatidylcholine (EYPC) and 1,2-dioleoyl-sn-glycero-3-phospho-L-serine (DOPS) in chloroform and POCs dissolved in chloroform. The lipids and POCs were added into a round bottom flask. Chloroform, diethyl ether, and (4-(2-hydroxyethyl)-1-piperazineethanesulfonic acid) (HEPES) buffer were then added in the same flask in a volume ratio of 2:1:1. The mixture was kept under dry argon. Subsequently, the mixture was sonicated at 0–4 °C until a homogeneous water-in-oil mixture was obtained. The organic solvents were then removed under a reduced pressure using a rotary evaporator (178 rpm, 45 °C, in the air). As the organic solvents were being evaporated, a gel-like form of the liposome sample was observed. Once the organic solvents were mostly removed, the sample became less viscous. The process was left to run until most of the organic solvents were removed and a homogeneous translucent liposome solution was obtained.

**Liposome preparation and confocal fluorescence microscopy**. Liposomes were prepared using egg-yolk phosphatidylcholine (EYPC) and 1,2-dioleoyl-sn-glycero-3-phospho-L-serine (DOPS) in a mole ratio of 4:1 using the reverse-phase method. EYPC (79.2 μL, 10 mg mL$^{-1}$)/DOPS (20.8 μL, 10 mg mL$^{-1}$) in chloroform and CC5 (0.03 fmCLR) or CC19 (0.07 fmCLR) dissolved in chloroform were added to a round-bottom flask. Chloroform, diethyl ether, and HEPES buffer (10 mM, pH 7.0) were then added in the same flask in a volume ratio of 2:1:1. The mixture was kept under dry argon. Subsequently, the mixture was sonicated at 0–4 °C until a homogeneous water-in-oil mixture was obtained. The organic solvents were then removed under a reduced pressure using a rotary evaporator (178 rpm, 45 °C, in the air). The liposome solution (20 μL) was spread between two glass coverslips and observed using a Nikon A1$^+$ confocal laser scanning microscope through an Apo ×60 Oil λS DIC N2 objective lens with a laser wavelength of 402 nm (laser line 405, laser power: 5.0, PMT HV: 100, A1 filter cube 450/50) to excite CC5 or CC19. NIS-Elements C software was used to acquire the images.

**Supported lipid bilayer (SLB) preparation for atomic force microscopy**. Liposomes with and without CC3 (0.03 fmCLR) were prepared using egg-yolk phosphatidylcholine (EYPC) and 1,2-dioleoyl-sn-glycero-3-phospho-L-serine (DOPS) in a mole ratio of 4:1 using the reverse phase method. EYPC (79.2 μL, 10 mg mL$^{-1}$)/DOPS (20.8 μL, 10 mg mL$^{-1}$) in chloroform and CC3 (0.03 fmCLR) dissolved in chloroform were added to a round-bottom flask. Chloroform, diethyl ether, and HEPES buffer (100 mM NaCl, 10 mM HEPES, pH 7.0) were then added in the same flask in a volume ratio of 2:1:1. The mixture was kept under dry argon. Subsequently, the mixture was sonicated at 0–4 °C until a homogeneous water-in-oil mixture was obtained. The organic solvents were then removed under a reduced pressure using a rotary evaporator (178 rpm, 45 °C, in air). Liposome samples were extruded through a hand-held extruder with 0.2 μm track-etched polycarbonate membrane for 21 times to obtain monodisperse, unilamellar vesicles and diluted to a final lipid concentration of 0.5 mg mL$^{-1}$ in the buffer (100 mM NaCl, 10 mM HEPES, pH 7.0). Freshly cleaved mica sheet (1 cm × 1 cm) was prepared beforehand and preheated on hot plate to 55 °C. Liposome solutions (200 μL) and calcium chloride (1 M, 6 μL) were gently add on the mica sheet and incubate at 55 °C for 15 min. Buffer (10 mM HEPES, pH 7.0) were added whenever needed to ensure the SLB samples did not dry up. The SLB samples were then washed with 1 mL buffer (10 mM HEPES, pH 7.0) for five times. SLB samples for liquid AFM were equilibrated in the buffer (10 mM HEPES) until observation under AFM in the same buffer. Liquid AFM was conducted on ParkSystems (Suwon, South Korea) using tapping mode and analyzed using Park Systems XEI 1.8. For solid AFM, SLB samples were washed with deionized water and dried in 55 °C oven for 10 min before the experiment. Solid atomic force microscopy (AFM) was conducted on Bruker Dimension ICON with Nanoscope V controller using tapping mode. The solid AFM data were processed with Nanoscope 9.7 and NanoScope Analysis 2.0.

**Neutron scattering measurements**. CC3 and CC19 crystals of one gram each were washed three times with diethyl ether and three times with ethanol. The crystals were dried under vacuum over 3 days to ensure good removal of residual solvent molecules. The dried CC3 and CC19 crystals were wetted with deuterium oxide (20 mL). The crystal suspensions were stirred for a week at room temperature to ensure thorough wetting of the crystals. Note that $D_2O$ was used to avoid the large incoherent neutron scattering background that would be produced by $H_2O$. Neutron powder diffraction data were collected using the BT-1 neutron powder diffractometer at the National Institute of Standards and Technology (NIST) Center for Neutron Research. A Ge(311) monochromator with $\lambda = 2.0787(2)$ Å was used. Wet powder samples were loaded into vanadium cans and sealed for the measurement. Due to the hydrophobicity of CC3 and CC19, there exists excess deuterium oxide within the samples, which cannot be removed without drying the samples. Hence, the samples were both measured at room temperature, at which the excess water is in liquid form and does not interfere with the diffraction from the main crystalline phase. Rietveld structural refinements were performed on the neutron diffraction data using the GSAS package. Multiple D sites with partial occupancies were used to model the $D_2O$ molecules within the structure, to account for their possible orientational disorder. Crystal data: (a) CC3: $C_{72}H_{84}D_{18\cdot48}N_{12}O_{9\cdot24}$, $M = 1302.50$, space group F41 3 2, cell parameters $a = b = c = 25.2235(9)$ Å. (b) CC19: $C_{288}H_{336}D_{81.02}N_{48}O_{56.53}$, $M = 5537.33$, space group F 41 3 2, cell parameters $a = b = c = 25.0543(9)$ Å.

**Preparation of lipid vesicles for water permeability measurements**. Liposomes were prepared using egg-yolk phosphatidylcholine (EYPC) and 1,2-dioleoyl-*sn*-glycero-3-phospho-L-serine (DOPS) in a mole ratio of 4:1 using the reverse-phase method. EYPC (79.2 μL, 10 mg mL$^{-1}$)/DOPS (20.8 μL, 10 mg mL$^{-1}$) in chloroform and POCs (various fmCLRs) dissolved in chloroform were added to a round-bottom flask. Chloroform, diethyl ether, and HEPES buffer (HEPES (10 mM), D(+) sucrose (200 mM), pH 7.0) were then added in the same flask in a volume ratio of 2:1:1. The mixture was kept under dry argon. Subsequently, the mixture was sonicated at 0–4 °C until a homogeneous water-in-oil mixture was obtained. The organic solvents were then removed under a reduced pressure using a rotary evaporator (178 rpm, 45 °C, in the air). The liposomes obtained were extruded through a hand-held extruder with 0.2 μm or 0.1 μm track-etched polycarbonate membrane for 21 times to obtain monodisperse, unilamellar vesicles. The excess buffer solution was added to make 0.5 mg lipids per mL stock solution. The size of the resulting liposomes was characterized by dynamic light scattering using a NanoBrook ZetaPlus particle electrophoresis system (Brookhaven Instruments).

**Water permeability measurements**. The water permeability measurements were conducted with a stopped-flow dynamic light scattering apparatus (Chirascan, Applied Photophysics). During the liposome-shrinking experiment, a hypertonic solution of HEPES buffer (10 mM, pH 7.0) with D(+)-sucrose (600 mM) was rapidly mixed with the liposome sample at 25 °C. This induced an outward-directed osmotic gradient, causing water to flow out from the liposome to the surrounding with an increased light-scattering signal at 90°. The change in liposome size is reflected by light scattering which was recorded at a wavelength of 600 nm and an angle of 90°. The analysis was carried out for 2 s such that the system reached equilibrium during this time and the graph plateaued. According to Rayleigh–Gans–Debye theory, the curve can be fitted by a double exponential fit function, with the $k_2$ being positively correlated with fmCLRs from 0 to 0.05. The exponential coefficient $k_2$ was then used to calculate the osmotic permeability ($P_f$) using the following equation,

$$P_f = \frac{k_2}{\frac{S}{V_0} \cdot V_w \cdot \Delta_{osm}}, \quad (1)$$

where $k_2$ (s$^{-1}$) is the exponential coefficient describing the change in light scattering during shrinkage of vesicles; $S$ and $V_0$ are initial surface area and volume of the vesicles, respectively, calculated from the mean diameter obtained from dynamic light scattering experiments; $V_w$ is the molar volume of water (0.018 L mol$^{-1}$); $\Delta_{osm}$ is the osmolarity difference, which was estimated to be 0.2 Osm L$^{-1}$ for 600 mM D (+)-sucrose hypertonic osmolyte. The corrected water permeabilities proposed by Pohl and co-workers[31,32] were calculated as follows,

$$P_f = \frac{k_2}{\frac{S}{V_0} \cdot V_w} \times \frac{c_{in,0} + c_{out}}{2c_{out}^2}, \quad (2)$$

where $k_2$ (s$^{-1}$) is the exponential coefficient describing the change in light scattering during shrinkage of vesicles; $S$ and $V_0$ are initial surface area and volume of the vesicles, respectively, calculated from the mean diameter obtained from dynamic light scattering experiments; $V_w$ is the molar volume of water (0.018 L mol$^{-1}$); $c_{in,0}$ is the intravesicular osmolyte concentration at $t = 0$, which is 0.2 Osm L$^{-1}$; $c_{out}$ is the extravesicular osmolyte concentration, which was estimated to be 0.4 Osm L$^{-1}$ for 600 mM D(+)-sucrose hypertonic osmolyte.

**Calculation of single-nanoaggregate permeability of POC**. The calculation was adapted from the previously reported method[8]. Each POC nanoaggregate consists of 17 discrete POC molecules and is assumed to have a cross-sectional area of 22.73 nm$^2$ (CC3, FT-RCC3, CC19) or 104.04 nm$^2$ (CC5). A lipid molecule has a cross-

sectional area of 0.35 nm$^2$ and the lipid bilayer has a thickness of ca. 5 nm. The 'unit area' was calculated based on one POC nanoaggregate and a variable number of lipid molecules depending on the feed mole ratio. The number of 'unit areas' is equivalent to the number of POC aggregates per liposome ($N$), which is given by

$$N = \frac{A_{Total}}{A_{Unit}} = \frac{2\pi r^2 + 2\pi(r-5)^2}{A_{POC} + \left(\frac{1-x}{x}\right)(0.35)}, \quad (3)$$

where $A_{Total}$ is the total surface area of a liposome; $A_{Unit}$ is the "unit area"; $A_{POC}$ is the cross-sectional area of POC; $x$ is the corrected fmCLR considering each nanoaggregate is made up of 17 POCs. It is important to note that the number of "unit area" needs to be halved in consideration of POC aggregates as transmembrane channels. The number of POC aggregates per liposome can then be used to calculate the single-nanoaggregate permeability ($P_a$) following Eq. (4). We adopted the single-channel permeability to calculate the single-nanoaggregate permeability ($P_a$),

$$P_a = \frac{(P_f) \times A}{N}, \quad (4)$$

where $P_f$ is the permeability; $A$ is the surface area of liposomes, and $N$ is the number of POC nanoaggregates incorporated in the liposome.

**Liposome preparation for ion selectivity measurements**. Liposomes were prepared using the reverse-phase method. DOPC in chloroform (100 μL, 10 mg mL$^{-1}$) and POCs (various fmCLRs) dissolved in chloroform were added to a round-bottom flask. Chloroform, diethyl ether, and HEPES buffer (pyranine (5 mM), HEPES (10 mM), sodium chloride (100 mM), pH 7.0) were then added in the same flask in a volume ratio of 2:1:1. For anion transport, HEPES buffer (pyranine (5 mM), HEPES (10 mM), pH 7.0) was used without the addition of inorganic salts. The mixture was kept under dry argon. Subsequently, the mixture was sonicated at 0–4 °C until a homogeneous water-in-oil mixture was obtained. The organic solvents were then removed under a reduced pressure using a rotary evaporator (178 rpm, 45 °C, in the air). Six rounds of freeze-thaw were conducted: the lipid vesicle sample solution was rapidly frozen in liquid nitrogen for 40 s and then thawed slowly in 45 °C water. The liposomes obtained were extruded through a hand-held extruder with a 0.2-μm-track-etched membrane for 21 times to obtain mono-disperse, unilamellar vesicles. Lastly, excess pyranine outside of liposomes in the surrounding HEPES buffer was removed by running through a 5 mL of Sephadex G-50 gel filtration column to give 0.5 mg DOPC per mL suspension.

**Cation selectivity studies by pyranine assay**. Liposome solution (80 μL, 0.5 mg DOPC per mL) and HEPES buffer (1920 μL, HEPES (10 mM), MCl (100 mM), pH 7.0, where M = Li$^+$, Na$^+$, K$^+$, Rb$^+$, or Cs$^+$) were placed in a fluorescence cell. Fluorescence spectra were collected at room temperature on a Photo Technology International/QuantaMaster (PTI/QM, USA) spectrometer. The temperature was set at 25 °C and the mixture was stirred for ~2 min in order to reach the set temperature. To the stirred solution, NaOH (20 μL, 0.5 M) was added at 50 s to induce a basic pulse and create a pH gradient across the lipid bilayer. Proton efflux from the vesicles and the charge are compensated with the influx of cations. For the gramicidin A control experiment, gramicidin A dissolved in dimethyl sulfoxide (265 mM) was added to the cuvette at 100 s. Triton X-100 (50 μL, 0.5 M) was added at 400 s to lyse the liposomes to release all pyranine and induce a maximum increase in intensity. The total experiment time was set at 400 s. Fluorescence measurements were carried out at excitation wavelengths of 403 and 460 nm, and an emission wavelength of 510 nm.

**Anion selectivity studies by pyranine assay**. Liposome solution (80 μL, 0.5 mg DOPC per mL) and HEPES buffer (1920 μL, HEPES (25 mM), pH 7.0) were placed in a fluorescence cell. Fluorescence spectra were collected at room temperature on a Photo Technology International/QuantaMaster (PTI/QM, USA) spectrometer. The temperature was set at 25 °C and the mixture was stirred for approximately 2 min in order to reach the set temperature. To the stirred solution, NaX (X = Cl$^-$, Br$^-$, I$^-$, 15 μL, 4 M) was added at 50 s to induce an ion pulse. Triton X-100 (50 μL, 0.5 M) was added at 500 s to lyse the liposomes to release all pyranine and induce a maximum increase in intensity. The total experiment time was set at 450 s. Fluorescence measurements were carried out at excitation wavelengths of 403 and 460 nm, and an emission wavelength of 510 nm.

**Simulation methodology**. Four POC-embedded lipid membranes were constructed to simulate water permeation, including CC3, CC19, small, and large CC5 nanoaggregates. Supplementary Fig. 5a illustrates a representative simulation system with the CC3-embedded lipid membrane. A CC3 crystal with a size of 4.73 × 4.73 × 4.73 nm$^3$ was inserted into a pre-equilibrated POPC (1-palmitoyl-2-oleoyl phosphatidylcholine) membrane. After energy minimization, the POC-embedded lipid membrane was further equilibrated by molecular dynamics (MD) simulation for 10 ns at 300 K and 1 bar. The temperature was controlled using the Nose–Hoover thermostat[39] with a relaxation time of 0.5 ps, whereas the semi-isotropic Parrinello–Rahman scheme was applied to maintain the pressure with a coupling constant of 5.0 ps. Water desalination was conducted at 300 K mimicking a forward osmosis (FO) process. A feed chamber with 2 M NaCl aqueous solution representing seawater and a permeate chamber with pure water were separated by the membrane. Two graphene layers were

placed outside the two chambers and exerted by atmospheric pressure (1 bar). In order to eliminate the effect of periodic images, a vacuum of 3 nm was added on each side. The POCs were modeled by the Optimized Potentials for Liquid Simulations all atom (OPLS-AA) force field[40]. The non-bonded potentials were also incorporated to describe the cage flexibility[19,20]. For the POPC lipids, the potential parameters were adopted from the Berger force filed reparametrized by Tieleman et al.[41]. Water was described using the TIP3P model[42], and the carbon atoms in graphene were described by the parameters used for carbon nanotubes. The cross interaction parameters were estimated by the Lorentz–Berthelot mixing rules[43].

Initially, each system was subjected to energy minimization using the steepest descent method. Then the velocities were assigned according to the Maxwell–Boltzmann distribution at 300 K. Finally, the production run was conducted in a canonical ensemble at 300 K for 50 ns. The atoms in the lipids and POC cages were allowed to fluctuate without position restraints. A cut-off distance of 1.2 nm was used to calculate the van der Waals interactions. Meanwhile, the particle-mesh Ewald method[44] was applied to evaluate the Coulombic interactions with a grid spacing of 0.12 nm and a fourth-order interpolation. The periodic boundary conditions were imposed in the three dimensions. The LINCS[45] and SETTLE[46] algorithms were used to constrain all the hydrogen-containing bonds and water molecules, respectively. The neighbor list was updated every ten steps. A time step of 2 fs was used and the trajectory was saved every 10 ps. For improved ensemble averages, ten independent simulations were conducted for the FO process with GROMACS v.5.0.6 package[47]. The analysis was conducted with GROMACS routines and locally written codes.

To calculate the binding energies of a single water molecule with CC3, CC19, and CC5 channels, i.e., calculated from potential energies, a POC channel with five cages was constructed from a crystal structure and the path in the channel was mapped out by Zeo++. Then, a single water molecule was placed at points along the channel center; at each point, the position of oxygen in the water was a constraint in $z$ axis while movable in $x$ and $y$ axes, and 1 ns MD simulation was performed; finally, the potential energy was averaged using the last 500 ps trajectory. During the simulation, the POC channel was fixed in three directions. $z$ is the direction along the channel center axis.

## Data availability

All data are available in the main text or the supplementary materials. The data that support the findings of this study are available from the corresponding authors upon reasonable request.

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

## Acknowledgements

This work was supported by the National Research Foundation Singapore (NRF-CRP17-2017-01, NRF2018-NRF-ANR007 POCEMON), the Ministry of Education - Singapore (MOE AcRF Tier 1 R-279-000-540-114, R-279-000-474-112, Tier 2 MOE2018-T2-2-148, MOE2019-T2-1-093), and the Agency for Science, Technology and Research (PSF 1521200078, IRG A1783c0015, and IAF-PP A1789a0024). We thank the staff from NUS Centre for Bioimaging Sciences (CBIS) Cryoelectron Microscopy Facility for their assistance with the cryogenic transmission electron microscopy. We thank the staff of Nanyang Technological University Central Environmental Science and Engineering Laboratory (CESEL) for their assistance with testing and analysis of liquid atomic force microscopy. We also thank Dr. Mihail Barboiu from Institut Européen des Membranes for confirming our ratiometric measurements. Finally, we would like to thank Ms. Fengyin Wu for aiding with image editing

## Author contributions

D.Z. (Dan Zhao) formulated and supervised the project. Y.D.Y. performed synthesis and purification of Pd@RCC3, CC1, CC3, CC5, and CC19; designed liposome reconstitution experiments; performed ratiometric measurements, voltage-clamping ion transport studies, stopped-flow permeability studies, cryo-TEM analyses, TEM analyses, EDX-mapping analyses, MS analyses, PXRD analyses, UV-Vis spectrometry analyses, contact angle analyses, liquid AFM preparation, solid AFM preparation and analyses, and FT-IR analyses. J.D. performed the synthesis and purification of RCC3 and FT-RCC3. J.L., D.Z. (Daohui Zhao), and J.J. performed the molecular simulations. H.W. and W.Z. performed the neutron powder diffraction measurements and data refinement. Y.W.T. provided access to stopped-flow equipment. H.X.G. and Y.W.T. participated in the discussion of water permeability determination. Y.D.Y. and D.Z. (Dan Zhao) wrote the manuscript. All authors contributed to the data analysis, discussion, and manuscript revision.

## Competing interests

The authors declare no competing interests.
