## [Peer Review File · Nature Communications]

Reviewers' comments:

Reviewer #1 (Remarks to the Author):

The manuscript by Zhao, Jiang, and coworkers presents the synthesis and study of porous organic cage/liposome constructs for the purpose of precisely tuned synthetic water channels. The authors state the materials will be useful for potential water desalination applications given their judiciously tuned pore channel size and hydrophobicity. Overall, I think the work is novel and of high quality and should ultimately be published in Nature Communications. However, there are some areas where the manuscript could be modified for clarity. Experimentally, I think the authors have done their diligence and their progression through pore size, pore surface chemistry, and particle size supports their claims. Some specific comments are listed below:

- 1) Although I appreciate that the authors are calling these cages 0D because they are discrete, it will be helpful to the general reader to clarify this.
- 2) The authors should clarify the orientation-independent text. It can be confusing because some POCs that should be porous may not be because their porosity depends on the orientation of their pore windows. Here, I believe, the authors are calling them orientation-independent because they have 3-D pore connectivities. This should be clarified.
- 3) The size regimes at play should be clarified here (bilayer vs. cage, vs. particle).
- 4) The authors talk about window size but quote pore size values. This should be corrected or the window opening size should be listed.
- 5) Along the lines of size/scale, it is unclear how many cages are actually stacked up in the bilayer. Figure 1 makes it look like a 3 x 3 x 3 particle. Is this accurate? How can they be sure?
- 6) The authors should clarify porosity, or lack thereof, as it pertains to cage collapse/flexing vs. non-optimal cage-cage packing.
- 7) Figure 3 could be improved. The cage-D2O interactions aren't very clear.
- 8) It is not immediately clear what 17-POC and 75-POC are.
- 9) In general, what is the scalability of this approach in terms of cost of cages and incorporation into membranes.

If the authors make these clarifications/changes, the manuscript is publishable.

Reviewer #2 (Remarks to the Author):

In this manuscript, Zhao and coworkers described an interesting work. By using the porous organic cages, the selective and efficient water transport was achieved in the lipid bilayers. I support its publication after revision according to the following suggestions:

1. It was claimed that the POCs could allow selective and fast water permeation. However, I noticed that the water transport measurements were performed in a hypertonic solution containing D(+)-sucrose. The size of sucrose is too big to flux through the cages. Under this condition, only water can flux through the cages. By using this condition, all porous molecules might allow the transport of water. So, I suggest to measure the water permeability in the hypertonic solution containing salt, such as NaCl and KCl.
2. It was claimed that the POCs could reject ions. However, the ion permeation measurements were performed under the condition different from that for water permeability measurements. I suggest to measure ion permeability and water permeability with the same hypertonic salt solution.
3. The single-channel water permeability was provided in the text. How can the channel density be determined? This can be further described in the main text.
4. The synthetic water channels are supramolecular (self-assembled from multiple molecules). The single-channel permeability was calculated by ignoring the dynamic feature of the channel structures. So, the obtained single-channel permeability is not accurate.

Reviewer #3 (Remarks to the Author):

This manuscript describes the application of porous organic cages (POCs) for synthetic water channels. The authors tested 6 distinct POCs in liposomes for water permeability and ion rejection and found that the window size, structural rigidity, the stability of assembled structure (the authors called nanoaggregates) and hydrophobicity of POCs are determining factors for the high performance. The authors further confirmed these experimental findings with the aid of molecular dynamics simulations and tried to understand molecular-scale mechanism of water permeability. Overall, this research contains a high novelty from the viewpoints of originality (research idea), experimental evidences, mechanistic studies by simulation. I would like to support the publication of this manuscript in Nature Communications with high recommendation. However, there are several scientific issues to be addressed before the acceptance.

1. The data summarized in Fig.2a look quite interesting; the higher loading was necessary for FT-RCC3 and CC19 for initiating water permeability that for CC3 and CC5. And the authors explained this tendency by hydrophobicity of FT-RCC3 and hydrophilicity of CC19.

I am wondering the meaning of thresholds observed in this POC system ($f_m\text{CLR} = 0.01$ for CC3 and CC5 and $f_m\text{CLR} = \text{FT-RCC3 and CC19}$). If we assume the formation of channels across the membrane is very fast, this plot of water permeability vs $f_m\text{CLR}$ should become linear because the water permeability should increase with the increase of the numbers of channels existing in the membrane as seen in other literatures like refs 7 and 8. However, this is not the case here. The authors explained this difference by hydrophobicity and hydrophilicity of POCs, which would explain the kinetics of water transport but cannot explain the existence of thresholds. I guess that the existence of thresholds might indicate a cooperative effect. What we should assume is the formation of nanoaggregates in lipids; below the thresholds, the concentration of POCs might be too low to form

nanoaggregates across the membrane and did not transport water molecules. In this assumption, FT-RCC3 and CC19 were less favored to form channels compared to CC3 and CC5. I am wondering if the authors can simulate such formation dynamics in lipids. In their simulation, 17-POCs were embedded in lipids and the authors confirmed the stability of these aggregates in the timescale of water transport. What happens when the smaller numbers of POCs were introduced in lipids? I would like the authors to address this issue.

2. In Fig.2c, it would be better to describe 'the molar fraction of CC1 vs CC3' for the x-axis because the authors described the total fmCLR as 0.03 in the index. Readers would easily see the effect of CC1 once it is written in the molar fraction.

Reviewer #4 (Remarks to the Author):

This work from Zhao, Jiang and coworkers reports the application of porous organic cages as synthetic water channels and water and ion transport properties. Porous organic cages are an interesting class of microporous materials. While the idea of using porous organic cages as water channels is interesting and of broad interest, the experimental data provided in the manuscript are insufficient to support their claims. Particularly the authors did not perform in-depth study of the structure of porous organic cages in the liposomes. I do not recommend this manuscript for publication in Nature Communications. Specific comments and concerns are listed below.

1. The paper is very difficult to read because there are so many claims and arguments without providing experimental evidence or explanation. Particularly, the authors cited a lot of papers instead of giving direct experimental data to support their claims, making it difficult to judge whether the results of this work are valid.
2. Porous organic cages can be easily fabricated into thin film membranes with thickness down to nanometer scale. Previous work by Cooper and co-workers have demonstrated the formation of amorphous cage thin film membranes (*Advanced Materials*. 2016, 13, 2629-2637) as well as oriented crystalline cage layers with window-to-window packing (*Angew. Chem. Int. Ed.* 2017, 56, 9391). The authors could easily follow previous studies and prepare pure selective cage membranes and study the water permeability and salt rejection but they did not do so.
3. The authors did not study the water adsorption in cage solids or diffusion coefficient. The authors could have studied the water diffusion coefficient using well-known characterization techniques such as pulsed field gradient NMR, which has been widely used to study water dynamics in zeolites. They could also prepare pure cage membranes and study water permeation and salt diffusion. Alternatively, the sorption and diffusion of water can be investigated using other characterization techniques, such as quartz crystal microbalance. Did the authors study the ion adsorption and quantify the capacity of ions adsorbed in the cages?
4. The authors prepared liposomes embedded with POCs. Such method has been widely used in the water channel community and used for membranes incorporated with protein channels. However, such method is not feasible for large scale membrane manufacturing in real applications. If the authors use this method to study the fundamental properties, such as water and ion permeability, they need to prove that the method is reproducible, and the results obtained are accurate.
5. According to the method of preparation of Liposome and control samples with cages incorporated

into liposome for confocal fluorescence microscopy, cages were dissolved in chloroform solvent and solvent was removed by rotary evaporator, the resulting liposome solution was spread between glass cover slips. It is not clear how CC5 and CC19 molecules would aggregate in the layer. Are these cage molecules forming ordered crystals or amorphous solids? Nanoaggregate is a very misleading word. Without evidence to determine the crystallinity of cages in the liposome, how can the authors claim the formation of orientation-independent water channels? There is very limited evidence to support these claims. The authors should perform control experiments using amorphous cages, such as cages that cannot crystallize (Nature Communications volume 2, Article number: 207 (2011)).

6. Similarly, the preparation of lipid vesicles for water permeability measurements, CC3 was dissolved in chloroform and the organic solvents were removed in rotary evaporator. Similar question: how cage molecules pack in the liposomes? Are the authors positive that these nanoaggregates are thicker or thinner than the lipid bilayer (5 nm)?

7. This reviewer has serious concerns about the accuracy of the reported water permeability and ion permeability data. The authors measured the water permeability by embedding the cage molecules into lipid vesicles. It is well known that vesicles made of only lipids have also a significant water permeability due to the passive diffusion of water owing to the flipping motion of lipids. The authors determined the water permeability by comparing the permeability of the vesicles with and without POCs embedded into the bilayer. The authors did not report the values of water permeability of the passive water diffusion. Furthermore, the water transport through the lipid bilayers embedded with cages might be affected by the interactions and interface between the lipid molecules and cages. Did the authors consider the possibility of molecular scale defects between lipid molecules and cage solids that can lead to water transport pathways? High resolution atomic force microscopy might be useful to study the structure of these liposomes embedded with cages and whether defects could be formed. Without considering these factors that can influence the water transport, the authors cannot claim that the water transport through the channels in POCs.

8. The authors assume POC nanoaggregate in the lipid layer as a single channel and calculate the single channel permeability. This is probably not correct because the nanoaggregate of POC consists of intrinsic cavities and external cavities, and if the nanoaggregate is amorphous then the pore network structure would be disordered. In both cases, cage solids can form molecular defects. All of these factors were not considered by the authors.

9. The results of water permeability of POCs are questionable. In Fig. 2C, the data shows that the water permeability of liposomes with CC1 and CC3. The data is strange because the water permeability is roughly in the range of 200-250 $\mu\text{m/s}$ with CC1/lipid molar ratio increases from 0-0.02, but then suddenly drops to zero when the CC1 ratio is increased to 0.025. Why such significant change happens when the CC1/lipid ratio increases from 0.02 to 0.025? Would CC1 and CC3 molecules pack together into amorphous solids in one nanoaggregate? In Fig. S4, water permeability of liposomes incorporated with equal portion of various POCs and CC3 are very similar (in the range of 200-250 $\mu\text{m/s}$), this is also strange, if the POC molecules with different structure are loaded in the same portion with CC3, why the water permeability are similar?

10. The water contact angle results suggest that CC3, CC5, FT-CC3 are actually very hydrophobic. If these POCs present water channels and show high water permeability as the authors claimed, then the water should be absorbed and diffuse into the cage solids and water contact angle should dynamically change and decrease quickly during the measurements, did the authors observe such change during the tests? By the way, the measurement of water contact angle of CC3 (126.6 degree)

is certainly not accurate.

11. A lot of experimental data were not clearly explained. For example, the ratiometric measurement data shown in Fig. S3. It was not explained what those lines mean and how the lines could be related to ion transport. The data looks very noisy.

12. The authors can not claim that these water channels could be used for desalination applications, since they probably only got some preliminary data. They have not proven the feasibility of continuous selective membranes and tested the membranes in real filtration processes, such as reverse osmosis and forward osmosis or even dialysis. There is a long way to go to develop desalination membranes from these materials, which will require new membrane manufacturing techniques.

Reviewer #5 (Remarks to the Author):

Yuan et al. report an interesting, and potentially exciting, piece of work, introducing the concept of using porous organic cages as synthetic water channels. The work comprises extensive experimental and computational efforts, demonstrating the strong expertise of the groups involved. While I trust that the interpretation of the experimental results and the conclusions of the work are sound, being a computational chemist, I suggest that the editor takes the assessment of the experimental details by a more knowledgeable reviewer. Overall, the work is exciting, the manuscript is well written, and the findings are of high significance, hence I would like to recommend its publication. Below are some technical points that hopefully will improve the manuscript.

(1) Please specify what exactly the energy is (i.e., free energy, potential energy, or some other definition) as it is in Figure 4d. It seems to me that they are some sort of binding energy as the authors used interaction energy, but this needs to be clearly defined in the relevant method section. Also, how were the paths mapped out? Was a single water molecule placed on specific points on the path, with the energy evaluated without optimizing the geometry? Or were the energy profiles mapped out by free energy calculations, such as umbrella sampling or potential of mean force? The authors should also specify what Z refers to as in the title of the horizontal axis of Figure 4d.

(2) Figure 4g shows profiles of the channel radii along the water path in each POC structure. Please specify based on which structure of each system these radii were calculated. It may be worth calculating multiple such profiles for each system, i.e., using multiple MD snapshots, to probe if these water channels show different, dynamic distributions of the channel radii.

(3) Surely, the analyses in Figure 4d & g need to be performed on relevant structures extracted from the simulations of the POC nanoaggregates within the POPC lipid bilayers. Please the authors clarify.

(4) Please the authors comment on whether or not simulated water permeabilities are sensitive to or dependent on the relative size of the POC nanoaggregate to the size of the lipid bilayer. Do we expect a converging behaviour of the water permeability as the hybrid system's cage-to-lipid ratio goes to the two extrema, i.e., the non-permeable lipid layer and the permeability of a slab of purely the cage molecules.

(5) The designation of CC19 first appeared in Jiang, et al., *Angew. Chem. Int. Ed.* 2018, 57, 11228, which should be cited for easy referencing should the reader be interested.

Response to Reviewers' Comments

Reviewer #1 (Remarks to the Author):

The manuscript by Zhao, Jiang, and coworkers presents the synthesis and study of porous organic cage/liposome constructs for the purpose of precisely tuned synthetic water channels. The authors state the materials will be useful for potential water desalination applications given their judiciously tuned pore channel size and hydrophobicity. Overall, I think the work is novel and of high quality and should ultimately be published in Nature Communications. However, there are some areas where the manuscript could be modified for clarity. Experimentally, I think the authors have done their diligence and their progression through pore size, pore surface chemistry, and particle size supports their claims. Some specific comments are listed below:

1) Although I appreciate that the authors are calling these cages 0D because they are discrete, it will be helpful to the general reader to clarify this.

Response: We thank the reviewer for pointing this out. We have added the explanation for POCs as zero-dimensional materials in the revised manuscript:

“Unlike other advanced porous materials such as metal–organic frameworks (MOF) or covalent organic frameworks (COFs) that occur as frameworks (three–dimensional, 3D), sheets (two–dimensional, 2D) and rods (one–dimensional, 1D), molecular cages can dissolve and exist as single molecular entity (zero–dimensional, 0D).”

2) The authors should clarify the orientation-independent text. It can be confusing because some POCs that should be porous may not be because their porosity depends on the orientation of their pore windows. Here, I believe, the authors are calling them orientation-independent because they have 3-D pore connectivities. This should be clarified.

Response: We are grateful of the reviewer for the apt understanding. We have added the suggested explanation in the revised manuscript:

“Most of such POCs can align window–to–window to form extended 3D pore networks consisting of internal cavities within each POC and external cavities between POCs where guest molecules can traverse (Figs. 1b–c) irrespective of the orientation of the POCs.”

3) The size regimes at play should be clarified here (bilayer vs. cage, vs. particle).

Response: We thank the reviewer for the suggestion. We have added the description of the size regime of the bilayer (ca. 5 nm), single POC (ca. 2 nm) and POC nanoaggregate (ca. 5 nm) in the revised manuscript:

“The presence of darkened objects within the lipid bilayer suggests that POCs are nanometer–scale in the lipid bilayer, which in turn controls the size of POC nanoaggregates within the bilayer thickness (ca. 5 nm). Each POC has a diameter of ca. 2 nm, which is too small to transverse the lipid bilayer. In order to prove this, we simulated an aggregate of POC containing three CC3 molecules in lipid bilayer (Video S1 and Supplementary Fig. S5b) and observed no water permeation through the POC aggregate. Therefore, we expect the POCs to form ca. 5 nm transmembrane nanoaggregates (Fig. 4a and Supplementary Fig. S5a) with short–range molecular ordering which is possible considering that symmetrical cages have high propensity to crystallize³⁰.”

4) The authors talk about window size but quote pore size values. This should be corrected or the window opening size should be listed.

Response: We are grateful of the reviewer for pointing the error in description. We have changed all the pore size to window size in the revised manuscript.

5) Along the lines of size/scale, it is unclear how many cages are actually stacked up in the bilayer. Figure 1 makes it look like a 3 x 3 x 3 particle. Is this accurate? How can they be sure?

Response: We thank the reviewer for the comment. Direct observation of the nanoscale POC in lipid bilayer is challenging. However, we have been able to deduce the size of the POC aggregate through the following indirect evidences: **(1)** There are thresholds in POC feed concentration versus water permeation observed during stopped-flow experiments, suggesting that there is a cooperative effect such that POCs need to aggregate to a certain size in order to traverse the membrane bilayer for water to permeate through. **(2)** This hypothesis is tested using simulation model showing no water passage in smaller POC nanoaggregate that is unable to traverse the lipid bilayer (Supplementary Fig. S5b in the revised Supplementary Information). **(3)** Previous studies have indicated that larger particles (> 10 nm) are difficult to be retained in the lipid bilayer as they will penetrate the lipid bilayer through membrane wrapping (Contini, C., Schneemilch, M., Gaisford, S. & Quirke, N. *J. Exp. Nanosci.* **2018**, *13*, 62). In addition, we have learnt from simulations that particles of sizes larger than 5 nm do not fit into the core of the lipid bilayer. On the other hand, particles of sizes less than 5 nm can fully nest within the bilayer and will not significantly disrupt the bilayer (Guo, Y., Terazzi, E., Seemann, R., Fleury, J. B. & Baulin, V. A. *Sci. Adv.* **2016**, *2*, e1600261). **(4)** Benefiting from the fluorescent property of two POCs, CC5 and CC19, we observed that these relatively hydrophobic materials preferentially reside in the lipid bilayer under confocal microscopy (Fig. 1f-g, Supplementary Fig. S1). **(5)** Both these fluorescent micrographs and our cryogenic TEM visualisation of liposomes containing POCs have shown perfectly formed liposomes without any membrane distortion compared to blank liposomes (Supplementary Fig. S2). **(6)** Furthermore, from the liquid AFM observation, there were no obvious protrusions in a CC3-incorporated supported lipid bilayer (SLB) compared to blank SLB (Supplementary Fig. S6). CC3-incorporated supported lipid bilayer appears to be rougher ($R_a = 0.657$ nm) compared to blank SLB ($R_a = 0.279$ nm) and freshly cleaved mica sheet ($R_a = 0.126$ nm). Hence, we concluded that the size of water-permeating POC nanoaggregates is controlled by the hydrophobic interaction with lipid bilayer and limited to around 5 nm, which corresponds to an aggregate with 17 POC molecules. In fact, we found that the experimental and simulated results are in fairly good agreement with each other.

“Fig. S6 | AFM images of blank liposome and liposome with CC3. 2D AFM images of blank supported lipid bilayer (SLB) (a) and CC3–incorporated (fmCLR 0.03) SLB (b) on mica sheet observed using tapping mode in buffer (10 mM HEPES). The same images were observed in 3D showing blank SLB (c) being smoother compared to CC3–incorporated (fmCLR 0.03) SLB (d) on mica sheet. Solid AFM was performed in tapping mode for blank SLB (e) and CC3–incorporated (fmCLR 0.03) SLB (f) on mica sheet. Some weak protrusions were observed in CC3–incorporated SLB and they are marked with white arrows. Larger CC3 nanoaggregates deposited on the SLB (circled in white) were also observed.

Note: Due to the smoothness of blank lipid bilayer. Solid SLB appears regularly patterned due to system noise.”

6) The authors should clarify porosity, or lack thereof, as it pertains to cage collapse/flexing vs. non-optimal cage-cage packing.

Response: We are grateful of the reviewer for the suggestion. We have added the suggested descriptions in the revised manuscript:

“Most of such POCs can align window-to-window to form extended 3D pore networks consisting of internal cavities within each POC and external cavities between POCs where guest molecules can traverse (Figs. 1b–c) irrespective of the orientation of the POCs.”

“It is well known that RCC3 has flexible amine bonds that can twist and collapse the cage when the cage is desolvated, resulting in a smaller or non-existent internal cavity²⁵.”

“Although CC1 has similar window opening size and rigidity as that of CC3, it differs from CC3 in its packing mode where the windows of each cage molecule are blocked by the arenes of its neighbours²³. This prevents the formation of an interconnected pore network within the CC1 packing despite still having both internal and external cavities.”

7) Figure 3 could be improved. The cage-D2O interactions aren't very clear.

Response: We are grateful of the reviewer for the suggestion. We have added lines as well as enlarged diagrams of the deuterium clusters in Fig. 3 to describe the possible hydrogen bonding between the deuterium oxides as well as between deuterium oxides and the POCs.

“Fig. 3 Water molecule distribution in POCs. Structures of D₂O in CC3 (a) and CC19 (b), determined from neutron powder diffraction measurements (Supplementary Fig. S9). Carbon, grey; nitrogen, blue; hydroxyl oxygen, pink. The oxygen atoms from deuterium oxide are presented in balls with various colours (i.e., red, orange, yellow, and light yellow) to indicate crystallographically independent D₂O species. Deuterium and hydrogen atoms are omitted for clarity. (c) Arrangement

of oxygen atoms (red) of D₂O in four closely packed CC19 molecules. Peripheral oxygen atoms are omitted for clarity. Crystallographic arrangement of D₂O in the internal cavity (d) and inter-POC space (e) of CC3 with possible hydrogen bonding between the D₂O molecules and with the POC. Nitrogen atoms from CC3 are represented as blue spheres. Note that orientational disorder of D₂O was expected at room temperature, and the D₂O molecules were modelled with multiple deuterium sites.”

8) It is not immediately clear what 17-POC and 75-POC are.

Response: We thank the reviewer for pointing this out. We have clarified this in our revised manuscript:

“Considering our observation that the nanoaggregates of POCs should traverse the bilayer with good channel connectivity, we inserted a nanocrystal consisting of 17 molecules of CC3 or CC19 (17-POC) into the lipid layer for the simulation. Instead, a larger 75-POC nanocrystal (i.e., 75 molecules of POC per nanocrystal) was used for CC5 as we observed the tendency of lipid tail invasion that can rapidly block the pores of the 17-POC nanocrystal of CC5 during simulation (Video S2 and Supplementary Fig. S5d).”

9) In general, what is the scalability of this approach in terms of cost of cages and incorporation into membranes.

Response: We thank the reviewer for this question. We believe POCs are cheaper and easier to synthesize compare to other biological membrane fillers such as aquaporins. For example, flow synthesis of POCs via dynamic covalent chemistry has been achieved at lab-scale (Briggs, M. E. et al. *Chem. Commun.* **2015**, *51*, 17390). Thereby, it is likely that large quantity of POCs can be produced via flow synthesis. Furthermore, the solution-processability of POCs gives them more flexibility to be compatible with the current desalination membrane fabrication process. Although this paper targets the fundamental understanding of POCs as water channels, we have briefly discussed the feasibility of POCs for scaled-up applications in the conclusion:

“These solution-processable molecules can potentially be homogeneously processed into composite materials such as membranes using facile engineering methods for desalination applications. Furthermore, the scaling-up ability of POC synthesis³⁶ can dramatically reduce the materials cost for larger scale applications.”

If the authors make these clarifications/changes, the manuscript is publishable.

Response: We thank the reviewer for this positive comment. We hope the above clarifications and changes are sufficient to address the review’s concerns.

Reviewer #2 (Remarks to the Author):

In this manuscript, Zhao and coworkers described an interesting work. By using the porous organic cages, the selective and efficient water transport was achieved in the lipid bilayers. I support its publication after revision according to the following suggestions:

1. It was claimed that the POCs could allow selective and fast water permeation. However, I noticed that the water transport measurements were performed in a hypertonic solution containing D(+)-sucrose. The size of sucrose is too big to flux through the cages. Under this condition, only water can flux through the cages. By using this condition, all porous molecules might allow the transport of water. So, I suggest to measure the water permeability in the hypertonic solution containing salt, such as NaCl and KCl.

Response: We thank the reviewer for the suggestion. We have tested the water permeability with sucrose being replaced by NaCl. These results are presented in Supplementary Fig. S7b and Table S4 in the revised Supplementary Information, and they are presented below:

“Fig. S7 | Water permeability of POCs. (a) Water permeability of liposomes with increasing feed loading of CC19 and FT–RCC3. (b) Water permeabilities of liposomes with various POCs (CC1 (fmCLR 0.03), CC3 (fmCLR0.03), RCC3 (fmCLR 0.03), FT–RCC3 (fmCLR 0.06), and CC19 (fmCLR 0.06)) when exposed to 0.6 M sucrose and sodium chloride environment. Stopped–flow light–scattering raw data for CC1 (c) and RCC3 (d) showing negligible water permeation improvement with increasing loading compared to blank liposome. (e) Stopped–flow light–scattering raw data of equal mass loading of CC3 (fmCLR 0.03) and ASPOC. (f) Powder X–ray diffraction of crystalline CC1 and CC3, ASPOC, as well as recrystallized equal molar CC1 and CC3. New peaks of recrystallized CC1/CC3 are marked with asterisk (*).”

“Table S4. Blank liposome permeability

Osmolyte	Filter size (nm)	Liposome size (nm)	k (s ⁻¹)	P_f (μm/s)	Corrected P_f (μm/s)
Sucrose	200	211.3 ± 9.8	12.1 ± 0.5	118.5 ± 4.5	44.5 ± 1.7
Sucrose	100	140.4 ± 19.8	25.9 ± 4.0	165.8 ± 14.4	62.2 ± 5.4
NaCl	200	200.6 ± 5.6	22.0 ± 1.0	203.9 ± 8.1	75.8 ± 1.8
NaCl	100	151.9 ± 2.3	39.8 ± 8.5	278.4 ± 42.6	104.4 ± 16.0

Note: Blank liposome permeabilities were tested with different liposome sizes as well as osmolytes. NaCl gave markedly higher permeabilities compared to sucrose. The shrinkage rates were obtained from single–exponential fitting of the stopped–flow data instead of double–exponential model. When fitting blank liposome data with double–exponential model, the two rates (k_1 , k_2) are either similar or only one gives meaningful data.”

We prepared blank liposomes of two different sizes by extruding them through 200 nm and 100 nm filters, respectively, both theoretically giving unilamellar vesicles. Comparing the water permeabilities of blank liposome under sucrose and NaCl reveals that lipid bilayers are more permeable when exposed to NaCl. In addition, CC3/CC5/CC19/FT–RCC3 also generally exhibited higher permeation in NaCl solutions, while CC1 and RCC3 did not show noticeable water permeation. It is important to note that the stopped–flow data were fitted to a double–exponential equation ($y = a_1e^{-k_1x} + a_2e^{-k_2x}$) with two rates (k_1 and k_2). One of the rates, k_1 , remained relatively constant (lipid bilayer permeability), while k_2 varied greatly from negative values (indicating either no channel or impermeable channels) to high positive values (indicating permeable channels). The permeabilities of POCs were extract from k_2 .

The higher permeabilities measured in NaCl solutions may be due to effects such as electrostatic interactions between salts and lipid bilayer (Böckmann, R. A., Hac, A., Heimbürg, T. & Grubmüller, H. *Biophys J.* **2003**, *85*, 1647; Gurtovenko, A. A. & Vattulainen, I. *J. Phys. Chem. B* **2008**, *112*, 1953). Another possible explanation concerns the effect of concentration polarization. According to Grzelakowski et. al., there exists two opposing forces of concentration polarization at the inner and external sides of the lipid bilayer when the liposome shrinks (Grzelakowski, M., Cherenet, M. F., Shen, Y.-X. & Kumar, M. *J. Membr. Sci.* **2015**, *479*, 223). When the liposome shrinks, the internal buffer concentration increases, which leads to concentration polarization. Simultaneously, the external side of the lipid bilayer experiences dilution effect due to the pure water exiting the liposome, resulting in dilutive concentration polarization. On the overall, the effective osmolarity gradient across the vesicle wall will be smaller compared to the bulk osmolarity difference (Fig. R1). However, with the lipid bilayer being more permeable towards NaCl, both the concentration polarization inside the liposome and the dilutive concentration polarization effect just outside the liposome will be smaller, resulting in a larger effective osmolarity gradient compared to the situation when sucrose was used as the osmolyte. This may have resulted in higher apparent POC permeabilities since more water molecules can exit when there is a larger osmolarity gradient.

Fig. R1. Effect of concentration polarization on water permeability of liposomes under different osmolytes (sucrose and NaCl). C_i and C_o are the osmolyte concentration on the inside and outside of the liposome, respectively. $(C_i)_a$ and $(C_o)_a$ are the apparent osmolyte concentrations near to the inner and external side of the lipid bilayer, respectively. $\Delta\pi_{\text{Bulk}}$ is the osmolarity difference in the bulk solutions inside and outside the liposome. $\Delta\pi_{\text{Effective}}$ is the actual osmolarity difference experienced by the liposome.

To avoid complications due to electrostatic interactions of ionic osmolyte as well as to compare with past literatures which employed large molecules (e.g., sucrose, pyranine) as osmolytes (Shen, Y. X. et al. *Proc. Natl. Acad. Sci. U.S.A.* **2015**, *112*, 9810; Licsandru, E. et al. *J. Am. Chem. Soc.* **2016**, *138*, 5403; Tunuguntla, R. H. et al. *Science* **2017**, *357*, 792), we shall refer to the water permeabilities of POCs in the presence of sucrose as the osmolyte in this study.

2. It was claimed that the POCs could reject ions. However, the ion permeation measurements were performed under the condition different from that for water permeability measurements. I suggest to measure ion permeability and water permeability with the same hypertonic salt solution.

Response: We thank the reviewer for the suggestion. We have tested NaCl permeability through CC5 using 600 mM NaCl extravesicular solution and 100 mM NaCl intravesicular solution. We chose CC5 as the model POC because of its larger pore size compared to that of CC3, which is more likely to allow ion permeation. However, we did not see any marked increase in ion permeability compared to the 100 mM NaCl extravesicular solution (Fig. R2).

Fig. R2. Ratiometric spectroscopy assay of CC5 at 100 mM or 600 mM NaCl extravesicular concentrations. Note that the intravesicular NaCl concentrations were kept at 100 mM in both tests.

3. The single-channel water permeability was provided in the text. How can the channel density be determined? This can be further described in the main text.

Response: We thank the reviewer for the suggestion. Single-channel or single-nanoaggregate water permeability was calculated with a few assumptions and have been included in our Methods section. We would like to highlight the rationale below:

“Calculation of single-nanoaggregate permeability of POC. The calculation was adapted from the previously reported method⁸. Each POC nanoaggregate consists of 17 discrete POC molecules and assumed to have a cross-sectional area of 22.73 nm² (CC3, FT-RCC3, CC19) or 104.04 nm² (CC5). A lipid molecule has a cross-sectional area of 0.35 nm² and the lipid bilayer has a thickness of ca. 5 nm. The ‘unit area’ was calculated based on one POC nanoaggregate and a variable number of lipid molecules depending on the feed mole ratio. The number of ‘unit areas’ is equivalent to the number of POC aggregates per liposome (N), which is given by

$$N = \frac{SA_{Total}}{SA_{Unit}} = \frac{2\pi r^2 + 2\pi(r-5)^2}{A_{POC} + \left(\frac{1-x}{x}\right)(0.35)} \quad [3]$$

where SA_{Total} is the total surface area of a liposome; SA_{Unit} is the ‘unit area’; A_{POC} is the cross-sectional area of POC; x is the corrected fmCLR considering each nanoaggregate is made up of 17 POCs. It is important to note that the number of ‘unit area’ needs to be halved in consideration of POC aggregates as transmembrane channels. The number of POC aggregates per liposome can then be used to calculate the single-nanoaggregate permeability (P_a) following equation [4]. We adopted the single-channel permeability to calculate the single-nanoaggregate permeability (P_a),

$$P_a = \frac{(P_f) \times SA}{N} \quad [4]$$

where P_f is the permeability; SA is the surface area of liposomes; and N is the number of POC nanoaggregates incorporated in the liposome.”

4. The synthetic water channels are supramolecular (self-assembled from multiple molecules). The single-channel permeability was calculated by ignoring the dynamic feature of the channel structures. So, the obtained single-channel permeability is not accurate.

Response: We thank the reviewer for the comment. We fully agree with the reviewer that a single-nanoaggregate permeability is not accurate. However, this calculation was done in order for easy comparison with the simulated model which only consists of a single POC nanoaggregate. Interestingly, we found that the experimental and simulated results are in fairly good agreement with each other.

Reviewer #3 (Remarks to the Author):

This manuscript describes the application of porous organic cages (POCs) for synthetic water channels. The authors tested 6 distinct POCs in liposomes for water permeability and ion rejection and found that the window size, structural rigidity, the stability of assembled structure (the authors called nanoaggregates) and hydrophobicity of POCs are determining factors for the high performance. The authors further confirmed these experimental findings with the aid of molecular dynamics simulations and tried to understand molecular-scale mechanism of water permeability. Overall, this research contains a high novelty from the viewpoints of originality (research idea), experimental evidences, mechanistic studies by simulation. I would like to support the publication of this manuscript in Nature Communications with high recommendation. However, there are several scientific issues to be addressed before the acceptance.

1. The data summarized in Fig.2a look quite interesting; the higher loading was necessary for FT-RCC3 and CC19 for initiating water permeability that for CC3 and CC5. And the authors explained this tendency by hydrophobicity of FT-RCC3 and hydrophilicity of CC19. I am wondering the meaning of thresholds observed in this POC system ($f_mCLR = 0.01$ for CC3 and CC5 and $f_mCLR = FT-RCC3$ and CC19). If we assume the formation of channels across the membrane is very fast, this plot of water permeability vs f_mCRL should become liner because the water permeability should increase with the increase of the numbers of channels exciting in the membrane as seen in other literatures like refs 7 and 8. However, this is not the case here. The authors explained this difference by hydrophobicity and hydrophilicity of POCs, which would explain the kinetics of water transport but cannot explain the existence of thresholds. I guess that the existence of thresholds might indicate a cooperative effect. What we should assume is the formation of nanoaggregates in lipids; below the thresholds, the concentration of POCs might be too low to form nanoaggregates across the membrane and did not transport water molecules. In this assumption, FT-RCC3 and CC19 were less favored to form channels compared to CC3 and CC5. I am wondering if the authors can simulate such formation dynamics in lipids.

Response: We thank the reviewer for this insightful comment. Indeed, the thresholds displayed is a result of cooperative effect such that a sufficient size of the nanoaggregate is needed to create a transmembrane channel. POC aggregates smaller than the lipid bilayer thickness are likely to be completely shielded from the surrounding waters by the lipid bilayer, as shown here in Fig. R3 below, and no water permeation was observed. Hence, below certain feed concentrations, we were unable to observe water transport as the developed POC nanoaggregates are unable to form transmembrane channels. However, the kinetics of water transport may affect the threshold as well. We agree with the reviewer that there are possibilities of CC19 and FT-RCC3 being less favoured to form water-permeating channels compared to CC3. However, the simulation work on the dynamic formation of POCs in lipid bilayers deserves an entire study on its own (Chan, H. & Král, P. *ACS Omega* **2018**, 3, 10631), and this will be conducted in future work to deepen the understanding of POC dynamics in lipid bilayer. We thank the reviewer again for the excellent suggestion.

2. In their simulation, 17-POCs were embedded in lipids and the authors confirmed the stability of these aggregates in the timescale of water transport. What happens when the smaller numbers of POCs were introduced in lipids? I would like the authors to address this issue.

Response: We thank the reviewer for this suggestion. We conducted an addition simulation with a smaller CC3 nanoaggregate (with 3 CC3 molecules) in the lipid membrane, as shown in Fig. R3 here and as Supplementary Fig. S5b in the revised Supplementary Information. For this case, no water permeation was observed (Video S1). This shows that a transmembrane structure is required for appreciable water permeation.

Fig. R3. Molecular dynamics (MD) simulation of possible water permeation through a CC3 nanoaggregate (3 CC3 molecules) in the lipid membrane. Note that no water permeation was observed in this case.

3. In Fig.2c, it would be better to describe ‘the molar fraction of CC1 vs CC3’ for the x-axis because the authors described the total fmCLR as 0.03 in the index. Readers would easily see the effect of CC1 once it is written in the molar fraction.

Response: We thank the reviewer for this comment. We have made the suggested correction in Fig. 2c in the revised manuscript.

“Fig. 2. Water permeability of liposomes embedded with POCs. (a) Permeabilities of POCs (CC3, FT-RCC3, CC5, and CC19) with loadings from 0 to 0.05 fmCLR measured under hypertonic conditions at

25 °C. Each data point shown here is the average of 3 replicates, and the error bars represent the standard deviations. (b) Representative stopped–flow traces from tests performed on liposomes embedded with CC3 of various fmCLRs. (c) Water permeabilities of liposomes with a combined fmCLR of 0.03 for CC1/CC3 mixture. Molar ratio of 0.0 refers to fmCLR of 0.03 for CC3 (no addition of CC1), and molar ratio of 1.0 refers to fmCLR of 0.03 for CC1 (no addition of CC3). (d) Single–channel water permeabilities of CC3 (this study) and synthetic channels with reported stopped–flow values^{4,8–10,12,13}. Pink colour indicates channels with total ion rejection while grey colour indicates channels with partial to no cation rejection. Notably, the CC3 water channel can push the limit of ion–rejecting channel diameter from 2.8 to 5.8 Å, as indicated by the arrow.”

Reviewer #4 (Remarks to the Author):

This work from Zhao, Jiang and coworkers reports the application of porous organic cages as synthetic water channels and water and ion transport properties. Porous organic cages are an interesting class of microporous materials. While the idea of using porous organic cages as water channels is interesting and of broad interest, the experimental data provided in the manuscript are insufficient to support their claims. Particularly the authors did not perform in-depth study of the structure of porous organic cages in the liposomes. I do not recommend this manuscript for publication in Nature Communications. Specific comments and concerns are listed below.

1. The paper is very difficult to read because there are so many claims and arguments without providing experimental evidence or explanation. Particularly, the authors cited a lot of papers instead of giving direct experimental data to support their claims, making it difficult to judge whether the results of this work are valid.

Response: We would appreciate if the reviewer could provide more detailed comments. For a brief clarification, this study was made on the basis of past progresses in the fields of POCs, ion channels, and water channels. Previous studies that have demonstrated the reversible uptake of water (Hasell, T., Schmidtman, M., Stone, C. A., Smith, M. W. & Cooper, A. I. *Chem. Commun.* **2012**, *48*, 4689), three-dimensional transport in POC (Liu, M. et al. *Nat. Commun.* **2016**, *7*, 12750), as well as water permeation and salt rejection through simulation modelling (Kong, X. & Jiang, J. *Phys. Chem. Chem. Phys.* **2017**, *19*, 18178; Zhao, D., Liu, J. & Jiang, J. *J. Membr. Sci.* **2019**, *573*, 177), are valuable evidences that POCs are capable to function as an ideal water channel. In this study, we have first confirmed the previous studies on the presence of water in POCs through neutron diffraction study. Then, we have contributed to the field with our studies on experimentally determining water and salt permeation through POCs, and the structural properties that may affect the functionalities of the POCs. Our simulation studies provide a microscopic view of the water transport in POC channels. We have also further explored the structural properties of POCs on the water and salt permeation under osmotic pressure, determined the permeation patterns (wetting-dewetting) and the binding energy of water molecule translocation in POCs. We would be happy to answer to specific areas of concerns from the reviewer.

2. Porous organic cages can be easily fabricated into thin film membranes with thickness down to nanometer scale. Previous work by Cooper and co-workers have demonstrated the formation of amorphous cage thin film membranes (*Advanced Materials.* **2016, *13*, 2629-2637) as well as oriented crystalline cage layers with window-to-window packing (*Angew. Chem. Int. Ed.* **2017**, *56*, 9391). The authors could easily follow previous studies and prepare pure selective cage membranes and study the water permeability and salt rejection but they did not do so.**

Response: We thank the reviewer for the suggestion. According to the previous study, as suggested by the reviewer, we have tried to spin-coat POCs onto anodized aluminium oxide (AAO) substrates (Fig. S8). From a preliminary test, we noticed a few problems with the fabricated membranes: (1) the POC film is fragile such that gently removing water droplets from the surface will likely disturb the POC coating, (2) some POCs (e.g., CC5) that have very limited solubility in solvents resulting in very diluted doping solutions that makes spin-coating very difficult. In an effort to obtain appreciable areas of coating, we have coated CC5 five times. Advanced experimental designs are required for even coating of the POCs on AAO substrates and to ensure the integrity of the membrane when in contact with liquid. We are thankful of the suggestion from the reviewer and we believe that the suggested membrane fabrication deserves an entire study on its own and is out of the scope of this current study.

Fig. S8 | Contact angles of POCs. (a) CC1, (b) CC3, (c) RCC3, (d) FT-RCC3, (e) CC5, and (f) CC19 coated on anodized aluminium oxide (AAO) substrate. (g) Time trace of a water drop (0.2 μL) on AAO substrate spin-coated with CC3 under enclosed environment. (h–i) AAO substrate coated with CC19 observed under white light (h) and UV light (365 nm, i) respectively. (j–k) AAO substrate coated with CC5 observed under white light (j) and UV light (365 nm, k) respectively. Note that CC5 is difficult to be coated evenly due to its limited solubility in solvents that resulted in a very diluted doping solution. Therefore, the sample shown in (j–k) was obtained after 5 times coating.”

3. The authors did not study the water adsorption in cage solids or diffusion coefficient. The authors could have studied the water diffusion coefficient using well-known characterization techniques such as pulsed field gradient NMR, which has been widely used to study water dynamics in zeolites. They could also prepare pure cage membranes and study water permeation and salt diffusion. Alternatively, the sorption and diffusion of water can be investigated using other characterization techniques, such as quartz crystal microbalance. Did the authors study the ion adsorption and quantify the capacity of ions adsorbed in the cages?

Response: We thank the reviewer for the suggestion. In this study, we have introduced POCs as water channels and studied the water and ion transport of POCs in liposomes using well-established methods (stopped-flow for water permeability and fluorescence spectroscopy assay for ion transport) for a fair comparison with the reported water channels. For the bulk POC system, we have studied the water adsorption in cage solids using neutron diffraction study and observed water residence in CC3 and CC19 (Fig. 3 in the manuscript). The suggestions from the reviewer are very valuable and inspirational, each deserves a stand-alone study on their own. In future works, we will study the tracing of water and ion diffusion in the POCs using pulsed field gradient NMR (Beckert, S. et al. *J. Phys. Chem. C* **2013**, *117*, 24866), and water sorption and diffusion in pure cage membranes.

4. The authors prepared liposomes embedded with POCs. Such method has been widely used in the water channel community and used for membranes incorporated with protein channels. However, such method is not feasible for large scale membrane manufacturing in real applications. If the authors use this method to study the fundamental properties, such as water and ion permeability, they need to prove that the method is reproducible, and the results obtained are accurate.

Response: We thank the reviewer for the comment. Recently, the study of water channel community has been extended to synthetic water channels that are nano-sized and can potentially be scaled up for industrial applications. The benefit of water channel is the nano-scale size that can potentially be added to conventional polymeric membranes to afford thin, ultrapermeable desalination membranes (Elimelech, M. & Phillip, W. A. *Science* **2011**, *333*, 712). We believe that the reviewer was referring to reproducing the high water permeation obtained in this study in membrane applications. There have been reports of pure POC membranes for gas separation (Song, Q. et al. *Adv. Mater.* **2016**, *28*, 2629) as well as molecularly mixed membrane containing POCs for organic solvent nanofiltration (Zhu, G. et al. *Angew. Chem. Int. Ed.* **2019**, *58*, 2638) with appreciable permeance. Therefore, we believe the feasibility of POCs in real applications is a matter of engineering feats involving the methods of POC incorporation and processing, etc. Although these studies are not the scope of this project, we wish we could embark on them in our future works.

5. According to the method of preparation of Liposome and control samples with cages incorporated into liposome for confocal fluorescence microscopy, cages were dissolved in chloroform solvent and solvent was removed by rotary evaporator, the resulting liposome solution was spread between glass cover slips. It is not clear how CC5 and CC19 molecules would aggregate in the layer. Are these cage molecules forming ordered crystals or amorphous solids? Nanoaggregate is a very misleading word. Without evidence to determine the crystallinity of cages in the liposome, how can the authors claim the formation of orientation-independent water channels? There is very limited evidence to support these claims. The authors should perform control experiments using amorphous cages, such as cages that cannot crystallize (Nature Communications volume 2, Article number: 207 (2011)).

Response: We are grateful for the reviewer's suggestions. Due to the nanoscale size regime in a solution-based system, we were unable to confirm the crystallinity of the POC aggregates. As suggested by the reviewer, we have obtained amorphous CC1/CC3 (ASPOC) equimolar mixture and used it as a control to investigate the effect of crystallinity on water permeation (S. Jiang *et al*, *Nat. Commun.* **2011**, *2*, 207). We loaded similar mass ratio of ASPOC as fmCLR 0.03 CC3 into liposome to observe if the amorphous POC mixture (PXRD shown in Supplementary Fig. S7f) allow water permeation. The stopped-flow DLS result showed a similar kinetic as that of a blank liposome (Supplementary Fig. S7e). This shows that the presence of the ASPOC does not affect the water permeation kinetics of the liposome. In addition, another indirect evidence indicating that POC may

retain short-range molecular ordering is the lack of water permeance in CC1, which is porous but exhibits apparent lack of porosity when aggregating together in a window-to-arene fashion. Should CC1 aggregates be amorphous, we would have seen water permeation through them due to inefficient molecular packing that can create some interconnected networks (Kong, X. & Jiang, J. J. *Phys. Chem. C* **2018**, *122*, 1732). In addition, according to previous simulation studies, the crystallinity of the POC nanoaggregates does not affect their orientation-independency as 3D channel network can form in both crystalline and amorphous POC aggregates (Kong, X. & Jiang, J. J. *Phys. Chem. C* **2018**, *122*, 1732). Each POC is a molecular entity with multiple windows leading to the same cavity. When individual POCs align together, 3D pore networks are formed and result in an orientation-independent network that allows guest permeation from many faces, instead of only two faces in a 1D channel.

6. Similarly, the preparation of lipid vesicles for water permeability measurements, CC3 was dissolved in chloroform and the organic solvents were removed in rotary evaporator. Similar question: how cage molecules pack in the liposomes? Are the authors positive that these nanoaggregates are thicker or thinner than the lipid bilayer (5 nm)?

Response: We thank the reviewer for the comment. Direct observation of the nanoscale POC in lipid bilayer is challenging. However, we have been able to deduce the size of the POC aggregate through the following indirect evidences: **(1)** There are thresholds in POC feed concentration versus water permeation observed during stopped-flow experiments, suggesting that there is a cooperative effect such that POCs need to aggregate to a certain size in order to traverse the membrane bilayer for water to permeate through. **(2)** This hypothesis is tested using simulation model showing no water passage in smaller POC nanoaggregate that is unable to traverse the lipid bilayer (Supplementary Fig. S5b in the revised Supplementary Information). **(3)** Previous studies have indicated that larger particles (> 10 nm) are difficult to be retained in the lipid bilayer as they will penetrate the lipid bilayer through membrane wrapping (Contini, C., Schneemilch, M., Gaisford, S. & Quirke, N. *J. Exp. Nanosci.* **2018**, *13*, 62). In addition, we have learnt from simulations that particles of sizes larger than 5 nm do not fit into the core of the lipid bilayer. On the other hand, particles of sizes less than 5 nm can fully nest within the bilayer and will not significantly disrupt the bilayer (Guo, Y., Terazzi, E., Seemann, R., Fleury, J. B. & Baulin, V. A. *Sci. Adv.* **2016**, *2*, e1600261). **(4)** Benefiting from the fluorescent property of two POCs, CC5 and CC19, we observed that these relatively hydrophobic materials preferentially reside in the lipid bilayer under confocal microscopy (Fig. 1f-g, Supplementary Fig. S1). **(5)** Both these fluorescent micrographs and our cryogenic TEM visualisation of liposomes containing POCs have shown perfectly formed liposomes without any membrane distortion compared to blank liposomes (Supplementary Fig. S2). **(6)** Furthermore, from the liquid AFM observation, there were no obvious protrusions in a CC3-incorporated supported lipid bilayer (SLB) compared to blank SLB (Supplementary Fig. S6). CC3-incorporated supported lipid bilayer appears to be rougher ($R_a = 0.657$ nm) compared to blank SLB ($R_a = 0.279$ nm) and freshly cleaved mica sheet ($R_a = 0.126$ nm). Hence, we concluded that the size of water-permeating POC nanoaggregates is controlled by the hydrophobic interaction with lipid bilayer and limited to around 5 nm, which corresponds to an aggregate with 17 POC molecules. In fact, we found that the experimental and simulated results are in fairly good agreement with each other.

7. This reviewer has serious concerns about the accuracy of the reported water permeability and ion permeability data. The authors measured the water permeability by embedding the cage molecules into lipid vesicles. It is well known that vesicles made of only lipids have also a significant water permeability due to the passive diffusion of water owing to the flipping motion of lipids. The authors determined the water permeability by comparing the permeability of the vesicles with and without POCs embedded into the bilayer. The authors did not report the values of water permeability of the passive water diffusion. Furthermore, the water transport through

the lipid bilayers embedded with cages might be affected by the interactions and interface between the lipid molecules and cages. Did the authors consider the possibility of molecular scale defects between lipid molecules and cage solids that can lead to water transport pathways? High resolution atomic force microscopy might be useful to study the structure of these liposomes embedded with cages and whether defects could be formed. Without considering these factors that can influence the water transport, the authors cannot claim that the water transport through the channels in POCs.

Response: We thank the reviewer for the comments and suggestions. The reviewer is correct in stating that lipid bilayer has water permeation to a certain extent and hence passive water permeability of blank liposomes should be significant. We have obtained the permeability values of blank liposomes to ensure that our liposomes have good integrity (Supplementary Table S4). It is important to note that we fitted the stopped-flow data to a double-exponential equation ($y = a_1e^{-k_1x} + a_2e^{-k_2x}$) with two rates (k_1 and k_2). One of the rates, k_1 , remained relatively constant (lipid bilayer permeability), while k_2 varied greatly from negative (either no channel or impermeable channel) to high values (permeable channel). The permeabilities of POCs were extract from k_2 . Blank liposomes can be fitted with both double-exponential or single-exponential ($y = ae^{-kx}$) fitting since there is only one transport pathway. The blank liposome single-exponential k is similar to double-exponential k_1 .

“Table S4. Blank liposome permeability

Osmolyte	Filter size (nm)	Liposome size (nm)	k (s ⁻¹)	P_f (μm/s)	Corrected P_f (μm/s)
Sucrose	200	211.3 ± 9.8	12.1 ± 0.5	118.5 ± 4.5	44.5 ± 1.7
Sucrose	100	140.4 ± 19.8	25.9 ± 4.0	165.8 ± 14.4	62.2 ± 5.4
NaCl	200	200.6 ± 5.6	22.0 ± 1.0	203.9 ± 8.1	75.8 ± 1.8
NaCl	100	151.9 ± 2.3	39.8 ± 8.5	278.4 ± 42.6	104.4 ± 16.0

Note: Blank liposome permeabilities were tested with different liposome sizes as well as osmolytes. NaCl gave markedly higher permeabilities compared to sucrose. The shrinkage rates were obtained from single–exponential fitting of the stopped–flow data instead of double–exponential model. When fitting blank liposome data with double–exponential model, the two rates (k_1 , k_2) are either similar or only one gives meaningful data.”

Molecular scale defects between lipid bilayer and POCs are possible and exist for channels and particles according to their charge, polarity, and size. The POCs we have selected are hydrophobic, neutral, and have molecular sizes of ca. 2 – 3 nm. Therefore, the major effect will come from the size of POC nanoaggregates formed. Previous studies have indicated that larger particles (> 10 nm) are difficult to be retained in the lipid bilayer as they will penetrate the lipid bilayer through membrane wrapping (Contini, C., Schneemilch, M., Gaisford, S. & Quirke, N. *J. Exp. Nanosci.* **2018**, *13*, 62). In addition, we have learnt from simulations that particles of sizes larger than 5 nm do not fit into the core of the lipid bilayer. On the other hand, particles of sizes less than 5 nm can fully nest within the bilayer and will not significantly disrupt the bilayer (Guo, Y., Terazzi, E., Seemann, R., Fleury, J. B. & Baulin, V. A. *Sci. Adv.* **2016**, *2*, e1600261). From our cryogenic TEM visualisation of liposomes containing POCs, we did not observe any membrane distortion compared to blank liposomes (Supplementary Fig. S2). This shows that the water permeation is not due to deformation of the liposomes, but as a result of insertion of POCs. Furthermore, during our simulation studies, we observed that the lipid tails are highly dynamic and tend to extend into the pores of the POC nanoaggregates. This will potentially seal the interfacial gaps between the lipid bilayer and the POC nanoaggregates.

We have conducted liquid AFM in accordance to the reviewer's suggestion. Due to the small size of POCs, we did not observe clear structure of POCs in the lipid bilayer although a markedly higher roughness was observed in CC3-incorporated lipid bilayer compared to blank lipid bilayer (Supplementary Fig. S6). We also did not notice any visible defects in the lipid bilayer.

8. The authors assume POC nanoaggregate in the lipid layer as a single channel and calculate the single channel permeability. This is probably not correct because the nanoaggregate of POC consists of intrinsic cavities and external cavities, and if the nanoaggregate is amorphous then the pore network structure would be disordered. In both cases, cage solids can form molecular defects. All of these factors were not considered by the authors.

Response: We thank the reviewer for the comment. First, we would like to clarify that the "single-channel" permeability refers to the water transport through a single POC nanoaggregate. To avoid confusion, we have changed the term to "single-nanoaggregate" water permeability in the revised manuscript. We fully agree with the reviewer that a single-nanoaggregate permeability is not accurate. However, this calculation was done in order for easy comparison with the simulated model which only consists of a single POC nanoaggregate. Interestingly, we found that the experimental and simulated results are in fairly good agreement with each other.

9. The results of water permeability of POCs are questionable. In Fig. 2C, the data shows that the water permeability of liposomes with CC1 and CC3. The data is strange because the water permeability is roughly in the range of 200-250 $\mu\text{m/s}$ with CC1/lipid molar ratio increases from 0-0.02, but then suddenly drops to zero when the CC1 ratio is increased to 0.025. Why such significant change happens when the CC1/lipid ratio increases from 0.02 to 0.025? Would CC1 and CC3 molecules pack together into amorphous solids in one nanoaggregate? In Fig. S4, water permeability of liposomes incorporated with equal portion of various POCs and CC3 are very similar (in the range of 200-250 $\mu\text{m/s}$), this is also strange, if the POC molecules with different structure are loaded in the same portion with CC3, why the water permeability are similar?

Response: We thank the reviewer for the comments. We have conducted another repeat for mixing CC1 and CC3 of various ratios and added them to the revised manuscript (Fig. 2c). Similar trend was observed again as we observed similar slight decrease in water permeation when CC3 content decreased from fmCLR 0.03 to fmCLR 0.015 (or molar ratio of CC1/CC3 from 0.0 to 0.5). This permeability is similar to that of the permeability of CC3 from fmCLR 0.015 to fmCLR 0.03. This suggests that pure CC3 and CC1 aggregations were formed instead of co-crystals. We have set up a model experiment where we mixed equimolar CC1 and CC3 together in dichloromethane and then remove the solvent to let the mixture recrystallize. The PXRD pattern shows that the recrystallized CC1/CC3 mixture is a combination of CC1 and CC3 crystal peaks; however, there are new peaks that are not present in pure CC1 or CC3 PXRD data (Supplementary Fig. S7f). Therefore, CC1 and CC3 may form co-crystals but they largely aggregate within their own species when in higher concentrations. Hence, it is reasonable that the water permeability of CC1/CC3 is similar to CC3 since majority of water permeates through CC3. The sudden drops when CC3 loading falls below fmCLR 0.015 is due to CC3's threshold loading. Interestingly, at a molar ratio of CC1/CC3 of 0.67 (fmCLR 0.01 of CC3), where the concentration of CC3 has fallen short of the threshold loading, we still observed some water permeation. This may be due to the presence of CC1 which packs with CC3 to allow certain interconnected pores through either CC3 or inefficient packing. Similarly, this can explain the similar water permeation when 0.015 fmCLR CC3 is mixed with equimolar RCC3, FT-RCC3, CC5, or CC19. Recognising that these data may not add value to the paper, we have removed them from the Supplementary Information.

10. The water contact angle results suggest that CC3, CC5, FT-CC3 are actually very hydrophobic. If these POCs present water channels and show high water permeability as the authors claimed, then the water should be absorbed and diffuse into the cage solids and water contact angle should dynamically change and decrease quickly during the measurements, did the authors observe such change during the tests? By the way, the measurement of water contact angle of CC3 (126.6 degree) is certainly not accurate.

Response: We thank the reviewer for the comment and for pointing out the error. We have corrected the contact angle of CC3 in Supplementary Fig. S8b and Table R1. Indeed, we have noticed absorption of water into POC films coated on AAO substrate. We have performed an adsorption test for CC3. However, the time taken for droplet absorption in this case is an inaccurate way to gauge the water permeability of the POCs. This is due to uncertainties from factors such as the time taken to wet the POCs and evaporation of the droplet considering the small volume used to avoid long video time.

Table R1. Contact angle and water absorption of CC3-coated AAO

Sample	Contact angle (°)	Time taken for absorption of 0.2 µl water (s)
Bare AAO	-	< 1
CC3	93.7 ± 1.3	343.8 ± 47.8

Note: Time-based contact angle data were obtained using 0.2 µL water droplets under enclosed environment. Contact angles were obtained when the droplets are stabilized on top of the POC-coated AAO. Note that all POC films on AAO substrates were dried overnight under vacuum at room temperature before testing.

11. A lot of experimental data were not clearly explained. For example, the ratiometric measurement data shown in Fig. S3. It was not explained what those lines mean and how the lines could be related to ion transport. The data looks very noisy.

Response: We thank the reviewer for the comment and suggestion. It will be really helpful if the reviewer can specify the experimental data that have not been explained well and we will make the necessary clarifications in our paper. Here is a brief summary of the ratiometric technique, which has been widely adopted in biophysics to study the ion transport across lipid membranes.

We measured the cation and anion permeations through POCs using fluorescence spectroscopy assays with pyranine, which is a ratiometric fluorescence probe. Pyranine exhibits two different absorption wavelengths (460 nm when pyranine is protonated and 403 nm when deprotonated), thereby acting as a pH probe to indirectly measure the transport of alkali metal ions. During the test, an extraventricular base pulse (NaOH) was introduced to create a pH gradient across the lipid bilayer (at 50 s). This induces either proton efflux or OH⁻ influx into liposomes. This transmembrane ion translocation is compensated by cation influx. An increase in fluorescence intensity at the excitation of 460 nm will be observed when the pH inside the liposome increases with deprotonation of pyranine. This will increase the ratio between fluorescence intensities excited at 460 nm and 403 nm. Hence the fluorescence intensity ratio (I_{460}/I_{403}) tracks the degree of deprotonation of pyranine. With more cation influx, more pyranine will be deprotonated, leading to increasing fluorescence intensity ratio. We did not observe significant increase in the fluorescence intensity ratio for both blank liposome and liposomes with POCs, indicating the absence of cation permeation. Note that the initial spike in the fluorescence intensity ratio curve is due to the minimal pyranine outside of liposomes that was immediately deprotonated upon addition of NaOH. In the presence of ion channel, such as Gramicidin A that allows permeation of cations, the fluorescence intensity ratio will increase rapidly. At the end of the experiment, a detergent was added to completely destroy the

liposomes to release all pyranine, resulting in a sudden spike in fluorescence intensity ratio curve as all pyranine molecules are deprotonated. Anion transports can be observed using the same principle.

12. The authors can not claim that these water channels could be used for desalination applications, since they probably only got some preliminary data. They have not proven the feasibility of continuous selective membranes and tested the membranes in real filtration processes, such as reverse osmosis and forward osmosis or even dialysis. There is a long way to go to develop desalination membranes from these materials, which will require new membrane manufacturing techniques.

Response: We thank the reviewer for the comment. We agree with the reviewer that many engineering studies have to be conducted in order to select a method, such as pure POC membranes, mixed matrix membranes, or thin film nanocomposite membranes to achieve continuous selective membrane layers and test them in filtration setups. Similar to the water channel papers that have preceded this paper, the fundamental understanding of the transport behavior in channels is indispensable and engineering/membrane works would be the future directions.

Reviewer #5 (Remarks to the Author):

Yuan et al. report an interesting, and potentially exciting, piece of work, introducing the concept of using porous organic cages as synthetic water channels. The work comprises extensive experimental and computational efforts, demonstrating the strong expertise of the groups involved. While I trust that the interpretation of the experimental results and the conclusions of the work are sound, being a computational chemist, I suggest that the editor takes the assessment of the experimental details by a more knowledgeable reviewer. Overall, the work is exciting, the manuscript is well written, and the findings are of high significance, hence I would like to recommend its publication. Below are some technical points that hopefully will improve the manuscript.

(1) Please specify what exactly the energy is (i.e., free energy, potential energy, or some other definition) as it is in Figure 4d. It seems to me that they are some sort of binding energy as the authors used interaction energy, but this needs to be clearly defined in the relevant method section. Also, how were the paths mapped out? Was a single water molecule placed on specific points on the path, with the energy evaluated without optimizing the geometry? Or were the energy profiles mapped out by free energy calculations, such as umbrella sampling or potential of mean force? The authors should also specify what Z refers to as in the title of the horizontal axis of Figure 4d.

Response: We thank the reviewer for the questions. The energies in manuscript Fig. 4d are the binding energies of a single water molecule within CC3, CC19 and CC5 channels, i.e., calculated from potential energies (not free energies). For these calculations, a POC channel with 5 cages was constructed from a crystal structure and the path in the channel was mapped out by Zeo++. Then, a single water molecule was placed at points along the channel center; at each point, the position of oxygen in water was constraint in z-axis while movable in x- and y-axes, and 1 ns MD simulation was performed; finally, the potential energy was averaged using the last 500 ps trajectory. During the simulation, the POC channel was fixed in three directions. Z is the direction along the channel center axis. We have added the above details in the Method section.

(2) Figure 4g shows profiles of the channel radii along the water path in each POC structure. Please specify based on which structure of each system these radii were calculated. It may be worth calculating multiple such profiles for each system, i.e., using multiple MD snapshots, to probe if these water channels show different, dynamic distributions of the channel radii.

Response: We thank the reviewer for the comment. We have plotted the RMSD evolution of POCs versus time in Supplementary Fig. S5c. The results indicate that POC nanoaggregates remain rather stable in the lipid bilayer. Therefore, the crystal structures of POCs constructed from X-ray crystallography data were used to estimate the channel radii.

“Fig. S5 | Molecular dynamics (MD) simulation of water permeation. (a) A representative simulation system, in which a CC3 nanocrystal with a dimension of $4.73 \times 4.73 \times 4.73 \text{ nm}^3$ was embedded in the 1-palmitoyl-2-oleoyl phosphatidylcholine (POPC) lipid membrane. Two chambers (2 M NaCl aqueous solution on the left and pure water on the right) were separated by the membrane. Two graphene pistons were placed outside the chambers and exerted by atmospheric pressure (1 bar). Colour codes: CC3 crystal, white-cyan spheres; POPC, green; Na^+ , orange; Cl^- , blue; H_2O , red-white spheres; graphene layers, cyan. (b) A CC3 nanoaggregate (3 CC3 molecules) in the lipid membrane. Note that no water permeation was observed in this case. (c) Root-mean-square deviation (RMSD) evolution of POCs versus time. Note that CC3, CC19 and CC5 nanoaggregates remain stable inside

the lipid bilayer. (d) Illustration of the small CC5 nanoaggregate (17 cages) quickly blocked by lipid tails during simulation. (e) Wetting–dewetting profile in large CC5 nanoaggregate (75 cages). Insert: wetting–dewetting profile in small CC5 nanoaggregate (17 cages). (f) Top view of a large CC5 nanoaggregate. (g) Side view of a large CC5 nanoaggregate. (h–j) Simulation snapshots at various time intervals for CC3 nanoaggregate, CC19 nanoaggregate, and large CC5 nanoaggregate, respectively. Note that CC3 displays a wetting–dewetting transition while CC5 and CC19 do not.”

(3) Surely, the analyses in Figure 4d & g need to be performed on relevant structures extracted from the simulations of the POC nanoaggregates within the POPC lipid bilayers. Please the authors clarify.

Response: We thank the reviewer for the comment. As mentioned above in the previous question, the POC nanoaggregates are rather stable. Therefore, the analyses of Fig. 4d for the interaction energies in POC channels were based on the POC crystal structures constructed from X-ray crystallography data.

(4) Please the authors comment on whether or not simulated water permeabilities are sensitive to or dependent on the relative size of the POC nanoaggregate to the size of the lipid bilayer. Do we expect a converging behaviour of the water permeability as the hybrid system’s cage-to-lipid ratio goes to the two extrema, i.e., the non-permeable lipid layer and the permeability of a slab of purely the cage molecules.

Response: We thank the reviewer for the comment. If we normalize the water permeability by the membrane area, then as the reviewer has pointed out, the permeability is expected to depend on the cage-to-lipid ratio, from non-permeable to the case of pure POC crystal. In both the simulation and experiment of this work, the number of water molecules permeating through a POC channel was estimated and the number was not normalized by the lipid membrane surface, thus not depending on the cage-to-lipid ratio.

(5) The designation of CC19 first appeared in Jiang, et al., Angew. Chem. Int. Ed. 2018, 57, 11228, which should be cited for easy referencing should the reader be interested.

Response: We thank the reviewer for the suggestion. We have cited this paper as ref #5 in the revised Supplementary Information.

REVIEWERS' COMMENTS:

Reviewer #1 (Remarks to the Author):

The authors have satisfactorily addressed my concerns, and have seemingly addressed the concerns of the other reviewers. This manuscript should now be accepted.

Reviewer #2 (Remarks to the Author):

The manuscript has been fully revised according to all of the reviewers' suggestions. So, I support its publication.

Reviewer #3 (Remarks to the Author):

The authors answered the previous comments accordingly and even demonstrated quite some additional experiments to address the claims by all reviewers. I appreciate their efforts. Overall, the current form of manuscript is more convincing and I support its publication in Nature Communication as is.

Reviewer #4 (Remarks to the Author):

I was the original Reviewer 4 of this paper. I have reviewed the revised manuscript and the response to all reviewers' comments. The author have addressed most of the comments well. The authors have performed additional experiments, including new water permeability tests, water contact angles, blank liposome water permeability data, AFM imaging, etc. Although the structures of 5-nm sized nanoaggregates of POCs are still not clear, the authors have made great efforts and provided resonable explanation. This revised paper is suitable for publication.

One major comment:

Whilst this work is mainly fundamental work on water transport in POC-based channels, which may have broad implications, the authors try to propose the application of POC as high -permeability synthetic channels for water desalination. The introduction starts from seawater desalination and development of high-permeability artificial water channels. It is well acknowledged by the community that the ultra-high water permeability membranes would have minimal impact toward increased energy efficiency in RO desalination. Instead, improving the water-salt selectivity is more important for developing next-generation membranes to enhance the desalination process efficiency. These viewpoints have been highlighted by Elimelech and coworkers have been published. If the authors still want to focus on desalination, then it would be better to add a few sentences in the introduction (or conclusion) to highlight these viewpoints and cite the references (listed below), rather than focusing solely on high water permeability. This is quite important because readers of the paper would have a balanced view of the development of novel materials for membranes and the significance of this work (overemphasis of high water permeability is a bit

narrow).

References:

The relative insignificance of advanced materials in enhancing the energy efficiency of desalination technologies. *Energy Environ. Sci.*, 2020, 13, 1694.

Towards single-species selectivity of membranes with subnanometre pores. *Nature Nanotechnology* 15, 426–436(2020).

Reviewer #5 (Remarks to the Author):

The authors have fully addressed my comments on the original submission, and I do not have any further comments.

Response to Reviewers' Comments

Reviewer #1 (Remarks to the Author):

“The authors have satisfactorily addressed my concerns, and have seemingly addressed the concerns of the other reviewers. This manuscript should now be accepted.”

Response: We thank the reviewer's time and effort in reviewing our paper.

Reviewer #2 (Remarks to the Author):

“The manuscript has been fully revised according to all of the reviewers' suggestions. So, I support its publication.”

Response: We thank the reviewer's time and effort in reviewing our paper.

Reviewer #3 (Remarks to the Author):

“The authors answered the previous comments accordingly and even demonstrated quite some additional experiments to address the claims by all reviewers. I appreciate their efforts. Overall, the current form of manuscript is more convincing and I support its publication in Nature Communication as is.”

Response: We thank the reviewer's time and effort in reviewing our paper.

Reviewer #4 (Remarks to the Author):

“I was the original Reviewer 4 of this paper. I have reviewed the revised manuscript and the response to all reviewers' comments. The author have addressed most of the comments well. The authors have performed additional experiments, including new water permeability tests, water contact angles, blank liposome water permeability data, AFM imaging, etc. Although the structures of 5-nm sized nanoaggregates of POCs are still not clear, the authors have made great efforts and provided resonable explanation. This revised paper is suitable for publication.

One major comment:

Whilst this work is mainly fundamental work on water transport in POC-based channels, which may have broad implications, the authors try to propose the application of POC as high -permeability synthetic channels for water desalination. The introduction starts from seawater desalination and development of high-permeability artificial water channels. It is well acknowledged by the community that the ultra-high water permeability membranes would have minimal impact toward increased energy efficiency in RO desalination. Instead, improving the water-salt selectivity is more important for developing next-generation membranes to enhance the desalination process efficiency. These viewpoints have been highlighted by Elimelech and coworkers have been published. If the authors still want to focus on desalination, then it would be better to add a few sentences in the introduction (or conclusion) to highlight these viewpoints and cite the references (listed below), rather than focusing solely on high water permeability. This is quite important because readers of the paper would have a balanced view of the development of novel materials for membranes and the significance of this work

(overemphasis of high water permeability is a bit narrow).

References:

The relative insignificance of advanced materials in enhancing the energy efficiency of desalination technologies. *Energy Environ. Sci.*, 2020, 13, 1694.

Towards single-species selectivity of membranes with subnanometre pores. *Nature Nanotechnology* 15, 426–436(2020).”

Response: We thank the reviewer for the insightful comment. We have added the two references in the introduction as well as in the conclusion:

“As highlighted by Patel et al, purely increasing water permeability will only marginally reduce specific energy consumption¹⁵. Increasing water-solute selectivity, i.e. improving salt rejection, would be more effective at improving RO efficiency. Hence, water channels with both high water permeability and low or negligible ion permeation are favored.”

[Reference no. 15] Patel, S. K. et al. The relative insignificance of advanced materials in enhancing the energy efficiency of desalination technologies. *Energy Environ. Sci.* **13**, 1694–1710 (2020).

“Considering their easily tunable window size and chemical nature, POCs are possible candidates for more directed and precise water separations such as solute-solute separation that can potentially minimize desalination post-treatment³⁷.”

[Reference no. 37] Epsztein, R. DuChanois, R. M., Ritt, C. L., Noy, A. & Elimelech, M. Towards single-species selectivity of membranes with subnanometre pores. *Nat. Nanotechnol.* **15**, 426–436 (2020).

Reviewer #5 (Remarks to the Author):

“The authors have fully addressed my comments on the original submission, and I do not have any further comments.”

Response: We thank the reviewer’s time and effort in reviewing our paper.